# DEAS: DEtached value learning with Action Sequence for Scalable Offline RL

**Changyeon Kim**[1][*]  **Haeone Lee**[1]  **Younggyo Seo**[2]  **Kimin Lee**[1][†]  **Yuke Zhu**[3,4][†]

[1]KAIST  [2]UC Berkeley  [3]The University of Texas at Austin  [4]NVIDIA

## Abstract

Offline reinforcement learning (RL) presents an attractive paradigm for training intelligent agents without expensive online interactions. However, current approaches still struggle with complex, long-horizon sequential decision making. In this work, we introduce *DEtached value learning with Action Sequence* (DEAS), a simple yet effective offline RL framework that leverages action sequences for value learning. These temporally extended actions provide richer information than single-step actions, enabling reduction of the effective planning horizon by considering longer sequences at once. However, directly adopting such sequences in actor-critic algorithms introduces excessive value overestimation, which we address through detached value learning that steers value estimates toward in-distribution actions that achieve high returns in the offline dataset. We demonstrate that DEAS consistently outperforms baselines on complex, long-horizon tasks from OGBench and can be applied to enhance the performance of large-scale Vision-Language-Action models that predict action sequences, significantly boosting performance in both RoboCasa Kitchen simulation tasks and real-world manipulation tasks.

## 1 Introduction

Offline reinforcement learning (RL) (Lange et al., 2012; Levine et al., 2020) enables learning from static datasets without incurring online data collection risks, while circumventing the need for expensive expert demonstrations. However, existing methods primarily focus on short-horizon tasks with dense rewards (Yu et al., 2020; Fu et al., 2020; Gulcehre et al., 2020; Mandlekar et al., 2021) and struggle to scale to complex long-horizon scenarios. Recent attempts using large-scale architectures (Kumar et al., 2023a;b; Chebotar et al., 2023; Springenberg et al., 2024) show promise, but their effectiveness on complex tasks remains unexplored.

To address the need for long-horizon evaluation, recent work (Park et al., 2025a;b) has proposed challenging benchmarks for complex offline RL and demonstrated that reducing the effective planning horizon (i.e., shortening the time span over which the agent must plan) in both value and policy learning via $n$-step TD updates with high $n$ values and hierarchical policies is essential. However, these approaches rely on goal-conditioned RL with explicit, expert-provided goals, which are often unavailable in practice. For instance, high $n$ values in $n$-step TD updates introduce increased bias and bootstrap error in standard RL without explicit goal information (Tsitsiklis & Van Roy, 1996; Kearns & Singh, 2000; Sutton & Barto, 2018).

These limitations underscore the need for alternative approaches to horizon reduction (reducing the planning horizon) that work without explicit goal conditioning. One promising direction is leveraging action sequences, which have shown success in behavior cloning (Pomerleau, 1988) for capturing noisy, temporally-relevant distributions in expert demonstrations (Chi et al., 2023; Zhao et al., 2023). However, existing attempts to use action sequences for RL remain insufficient for achieving robust horizon reduction. Q-chunking (Li et al., 2025b) has explored the use of action sequences for RL, demonstrating their potential for temporally consistent exploration. However, introducing action sequences to standard actor-critic frameworks causes severe value overestimation (Seo & Abbeel, 2025) due to actors maximizing over potentially erroneous critic estimates with widely spanned action spaces. This problem is exacerbated in offline RL where distribution shift creates extrapolation errors (Kumar et al., 2019; Fujimoto et al., 2019; Kumar et al., 2020). While CQN-AS (Seo

---

[*]Work done while visiting The University of Texas at Austin.   **Project page:** https://changyeon.site/deas
[†]Equal advising.

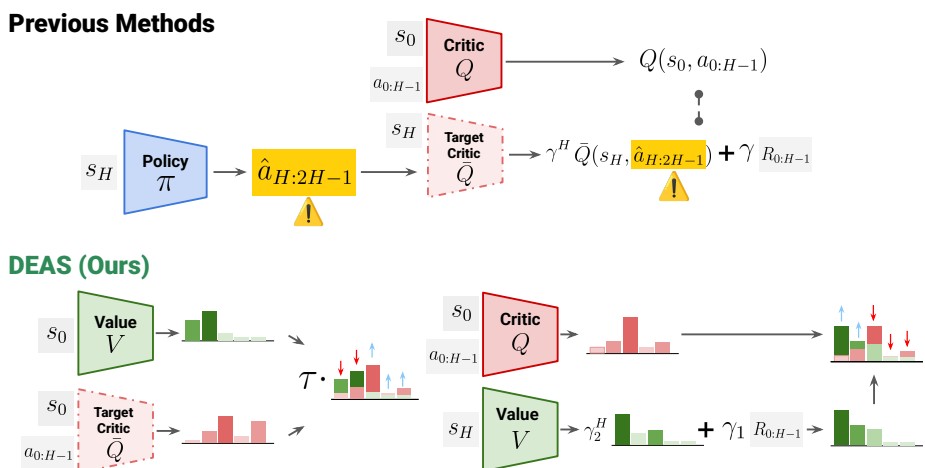

Figure 1: **Overview.** DEAS is an offline RL framework that learns from action sequences instead of single actions. Unlike previous methods that couple actor-critic training, our key insight is to train the critic separately from the policy (detached value learning) using action sequences, which enables stable learning while avoiding value overestimation. We further enhance stability by combining distributional RL objectives and using dual discount factors, which leads to additional improvement.

& Abbeel, 2025) proposes a value-only approach to avoid this issue, it introduces discretization errors that limit performance in complex tasks and cannot leverage expressive policy classes (Wang et al., 2023; Hansen-Estruch et al., 2023; Park et al., 2025c). For this reason, our research aims to develop methods that can leverage action sequences for horizon reduction while avoiding value overestimation and maintaining compatibility with expressive policy architectures.

**Our approach**   We present *DEtached value learning with Action Sequence* (DEAS), an offline RL framework that leverages action sequences for scalable value learning in complex tasks. Our method treats consecutive action timesteps as inputs to the value function to provide more expressive information than single-step actions. This design provides principled horizon reduction analogous to $n$-step TD updates with temporally extended actions, while action sequences offer richer information than single-step actions without requiring explicit goal conditioning. To address the value overestimation challenges inherent in learning value functions with action sequences in offline RL settings, we employ detached value learning (Kostrikov et al., 2022) that decouples critic training from the actor, biasing value estimates toward high-return actions present in the offline dataset. This method is appealing as it can be applied to any expressive policy architectures including large-scale Vision-Language-Action models (VLAs) without the hazard of value overestimation. Additionally, we propose to incorporate distributional RL (Farebrother et al., 2024) in value learning to mitigate instability from accumulated bias in multi-step returns.

We validate DEAS through comprehensive experiments on challenging long-horizon tasks from OGBench (Park et al., 2025a), where standard offline RL methods struggle to achieve meaningful success rates. Our method consistently outperforms all baselines, demonstrating its effectiveness on complex tasks. Additionally, we show that DEAS can be used to improve the performance of VLAs (Bjorck et al., 2025) in hard tasks from RoboCasa Kitchen (Nasiriany et al., 2024) and real-world manipulation tasks, which significantly improves performance compared to policies trained solely on expert demonstrations. These results demonstrate DEAS's practical applicability and potential for scaling offline RL to real-world scenarios.

**Contributions**   We highlight the key contributions of our paper below:

- We present DEAS: *DEtached value learning with Action Sequence*, a simple yet effective offline RL method that leverages action sequences for training critics and employs detached value learning with classification loss for stable training.

- We demonstrate that DEAS significantly outperforms baselines on complex, long-horizon tasks across 30 diverse scenarios in OGBench (Park et al., 2025a).

- We demonstrate that DEAS can enhance the performance of large-scale VLAs, achieving superior results on complex tasks from RoboCasa Kitchen (Nasiriany et al., 2024) and real-world manipulation tasks compared to policies trained solely on expert demonstrations.

## 2 RELATED WORK

**Offline reinforcement learning**   Offline RL focuses on learning policies from fixed datasets without further environment interaction (Levine et al., 2020). The primary challenge lies in the distributional shift between the behavior policy and the learned policy, which can result in value overestimation and suboptimal performance. Previous work has proposed various approaches including weighted regression (Peng et al., 2019; Nair et al., 2020; Wang et al., 2020), conservative regularization (Kumar et al., 2020), behavioral regularization (Fujimoto et al., 2019; Fujimoto & Gu, 2021; Tarasov et al., 2023; Park et al., 2025c), and in-sample distribution maximization (Kostrikov et al., 2022; Xu et al., 2023; Garg et al., 2023). Our method builds upon in-sample distribution maximization approaches, particularly IQL (Kostrikov et al., 2022), extending them to handle action sequences while maintaining stability by removing the critic update based on the actor's output. Furthermore, our method has the advantage of being adaptable to any policy extraction method for the final policy, making it more flexible and practical.

**BC/RL with action sequence**   Adopting action sequence has been actively investigated in both imitation learning and RL. Behavior cloning advances show that predicting action sequences captures temporal dependencies from expert demonstrations that single-step actions miss (Chi et al., 2023; Zhao et al., 2023; Black et al., 2025; Bjorck et al., 2025; Intelligence et al., 2025). Several works have introduced action sequences into RL (Li et al., 2024; Tian et al., 2025), with Q-Chunking (Li et al., 2025b) demonstrating incorporation into actor-critic frameworks in offline-to-online RL without policy class constraints. However, this approach faces fundamental challenges: expanded action spaces increase value overestimation risk, particularly in offline settings with limited data coverage (Kumar et al., 2019), yet this issue remains unaddressed. CQN-AS (Seo & Abbeel, 2025) circumvents this by removing the actor entirely, but introduces accumulating discretization errors that severely limit performance in complex tasks and prevent use of expressive policy classes (Wang et al., 2023; Park et al., 2025c). Our approach uniquely combines both paradigms: we leverage horizon reduction from action sequences while addressing value overestimation through detached value learning, enabling stable training with any policy architecture.

## 3 PRELIMINARIES

**Problem formulation**   We consider a Markov Decision Process (MDP) (Sutton & Barto, 2018) $\mathcal{M} = (\mathcal{S}, \mathcal{A}, p, R, \rho_0, \gamma)$, where $\mathcal{S}$ is the state space, $\mathcal{A}$ is the action space, $R(s, a) : \mathcal{S} \times \mathcal{A} \to \mathbb{R}$ is the reward function, $p(s'|s, a) : \mathcal{S} \times \mathcal{A} \to \Delta(\mathcal{S})$ is the transition function, $\rho_0$ is the initial state distribution, and $\gamma$ is the discount factor. In this paper, we focus on offline reinforcement learning, where we have access only to a static dataset $\mathcal{D} = \{\tau^i\}_{i=0}^{N}$ containing $N$ trajectories of fixed length $H$, where each trajectory $\tau^i = (s_0, a_0, r_0, \ldots, s_H, a_H, r_H)$ represents a sequence of states, actions, and rewards. The dataset is collected using a data collection policy $\pi_{\mathcal{D}} : \mathcal{S} \to \Delta(\mathcal{A})$, which may be unknown or suboptimal. Unlike online RL, we cannot interact with the environment during training. The objective is to learn a policy $\pi : \mathcal{S} \to \Delta(\mathcal{A})$ that maximizes the expected sum of discounted rewards $\mathbb{E}_{\rho_0, \pi, p} \left[ \sum_{t=0}^{\infty} \gamma^t R(s_t, a_t) \right]$ using only this fixed dataset.

**Temporal-difference (TD) learning**   Modern value-based RL approaches usually update a state-action value function $Q : \mathcal{S} \times \mathcal{A} \to \mathbb{R}$ to approximate the maximum discounted cumulative reward by bootstrapping from a successor state:

$$Q(s_t, a_t) \leftarrow Q(s_t, a_t) + \alpha \left[ r_t + \gamma \max_{a' \in \mathcal{A}} Q(s_{t+1}, a') - Q(s_t, a_t) \right].$$

This one-step update progressively aligns $Q$ with its Bellman target using sampled transitions. For reducing the bootstraping bias problem in long-horizon tasks, multi-step TD (Tsitsiklis & Van Roy, 1996; Kearns & Singh, 2000; Sutton & Barto, 2018) extends this idea by accumulating rewards over several steps before bootstrapping while retaining the same basic structure.

**Implicit Q Learning (IQL) (Kostrikov et al., 2022)** Instead of regularizing the critic with the actor output, IQL approximates the optimal critic to be maximized only in the region of action distributions present in the offline dataset with an in-sample expectile regression. Given a parameterized critic $Q(s_t, a_t; \theta)$, target critic $Q(s_t, a_t; \bar{\theta})$, and value network $V(s_t; \psi)$, the objective for value learning is defined as:

$$\mathcal{L}_V(\psi) = \mathbb{E}_{(s_t,a_t)\sim\mathcal{D}} \left[ L_2^\tau(\bar{Q}(s_t, a_t; \bar{\theta}) - V(s_t; \psi)) \right] \tag{1}$$

$$\mathcal{L}_Q(\theta) = \mathbb{E}_{(s_t,a_t,s_{t+1})\sim\mathcal{D}} \left[ (R(s_t, a_t) + \gamma V(s_{t+1}; \psi) - Q(s_t, a_t; \theta))^2 \right] \tag{2}$$

where $\mathcal{L}_2^\tau(u) = |\tau - \mathbb{1}(u < 0)|u^2$ is the expectile loss with expectile parameter $\tau \in [0, 1]$. By using $\tau > 0.5$, Equation (1) applies the higher weight on positive errors than negative errors, so that $V$ approximates an upper expectile of the in-distribution TD targets. This objective allows $V$ and $Q$ to be approximated only within the in-distribution region of actions.

## 4 METHOD

We propose DEtached value learning with Action Sequence (DEAS), an offline RL method that models action sequences for scalable value learning. Our approach consists of two key components: (1) a critic $Q(s_t, \mathbf{a}_t; \theta)$ that estimates expected returns for $H$-step action sequence $\mathbf{a}_t := a_{t:t+H-1}$ from state $s_t$ under the data collection policy $\pi_\mathcal{D}$, and (2) a flexible policy update mechanism applicable to any policy $\pi(\mathbf{a}_t; s_t, \phi)$ that outputs $H$-step action sequences. Section 4.1 describes how we extend temporal with action sequences, while Section 4.2 introduces how DEAS enables stable training through detached value learning, distributional RL, and dual discount factors. We provide pseudocode in Algorithm 1 and additional details in Appendix B.

### 4.1 TD LEARNING OVER ACTION SEQUENCES

Complex tasks require coordinated action sequences where each action's effectiveness depends on its context within the sequence. For instance, in OGBench `puzzle` or `cube` tasks, success depends on planning through multiple intermediate steps and maintaining consistent actions over extended periods. These temporal dependencies and hidden sub-tasks are not captured by current state representations, making it challenging for agents to learn effective policies. The challenge becomes even greater in goal-free settings, where agents must discover these sequential patterns from offline data without explicit goal instructions.

To address this challenge, we formalize temporally extended decisions by treating each fixed-length $H$-step action sequence as a single decision unit. Given the underlying MDP $\mathcal{M}$, we consider an state-action value function $Q(s, \mathbf{a})$ over H-step sequence of actions $\mathbf{a}_t := (a_t, a_{t+1}, \ldots, a_{t+H-1}) \in \mathcal{A}^H$. Executing $\mathbf{a}_t$ from state $s_t$ means applying the primitive actions in order, collecting the discounted return

$$\tilde{R}(s_t, \mathbf{a}_t, \gamma) := \mathbb{E}\left[ \sum_{k=0}^{H-1} \gamma^k R(s_{t+k}, a_{t+k}) \,\middle|\, s_t, \mathbf{a}_t \right], \tag{3}$$

and transitioning to a successor state $s_{t+H}$. Our TD updates are then performed directly on $Q(s_t, \mathbf{a}_t)$ using $\tilde{R}(s_t, \mathbf{a}_t, \gamma)$ as the multi-step target, i.e., standard TD learning applied to sequence actions in $\mathcal{A}^H$. This formulation is equivalent to considering decision-making at every $H$-th time step with temporally extended actions.

For updating $Q(s, \mathbf{a})$, we can use the following TD learning objective that extends standard Q-learning (Bradtke & Duff, 1994; Sutton et al., 1999):

$$Q(s_t, \mathbf{a}_t) \leftarrow Q(s_t, \mathbf{a}_t) + \alpha \left[ \tilde{R}(s_t, \mathbf{a}_t, \gamma_1) + \gamma_2^H \max_{o' \in \mathcal{A}^H} Q(s_{t+H}, o') - Q(s_t, \mathbf{a}_t) \right]$$

where $\gamma_1$ is used to construct the $H$-step return within a sequence, while $\gamma_2$ discounts across sequence-level decision points. This formulation aggregates rewards over $H$ steps and propagates value estimates across temporally extended transitions, achieving horizon reduction similar to $n$-step TD learning (Park et al., 2025b) while guaranteeing unbiased value estimates for the $H$-step rewards (Li et al., 2025b).

---

**Algorithm 1** DEAS

---

**Required**: Offline dataset $\mathcal{D}$, Support range for return $\mathbf{v}_{\min}, \mathbf{v}_{\max}$, number of bins $m$, discount factor $\gamma_1, \gamma_2$
Initialize parameters $\psi, \theta, \bar{\theta}, \phi$
**while** not converged **do**
    Sample batch $\{(s_t, \mathbf{a}_t, R_{t:t+H-1}, s_{t+H})\}$ from $\mathcal{D}$
    Compute the discounted return of $H$-step action sequence $\tilde{R}(s_t, \mathbf{a}_t, \gamma)$ using Equation (3)
    Compute $\bar{Q}(s, \mathbf{a}; \bar{\theta})$ and $V(s; \psi)$ using Equation (6)
    ▷ Update V Network
    Update $V(s; \psi)$ to minimize Equation (7) with $\bar{Q}(s, \mathbf{a}; \bar{\theta})$ and $V(s; \psi)$
    ▷ Update Q Network
    Update $Q(s, \mathbf{a}; \theta)$ to minimize Equation (8)
    ▷ Update Actor Network
    Update $\pi(s; \phi)$ with any type of policy extraction algorithms (e.g., BoN, DPG, AWR, etc.)
    Update $\bar{\theta} = (1 - \beta) \cdot \bar{\theta} + \beta \cdot \theta$
**return** $\pi(s)$

---

## 4.2 DEAS: DEtached value learning with Action Sequence

**Detached value learning for handling action sequence** Action sequences introduce challenges for value function approximation, as the expanded action space makes it harder for the critic to estimate Q-values accurately. Meanwhile, the actor can exploit regions where the critic makes prediction errors, leading to value overestimation and unstable learning (Seo & Abbeel, 2025). To address this, we adopt detached value learning (Kostrikov et al., 2022; Xu et al., 2023; Garg et al., 2023) that decouples actor and critic training, introducing a critic $Q(s_t, \mathbf{a}_t; \theta)$ and a value $V(s_t; \psi)$ networks following IQL (Kostrikov et al., 2022):

$$\mathcal{L}_V(\psi) = \mathbb{E}_{(s_t, \mathbf{a}_t) \sim \mathcal{D}} \left[ L_2^\tau(\bar{Q}(s_t, \mathbf{a}_t; \bar{\theta}) - V(s_t; \psi)) \right] \tag{4}$$

$$\mathcal{L}_Q(\theta) = \mathbb{E}_{(s_t, \mathbf{a}_t) \sim \mathcal{D}} \left[ (\tilde{R}(s_t, \mathbf{a}_t, \gamma_1) + \gamma_2^H V(s_{t+H}; \psi) - Q(s_t, \mathbf{a}_t; \theta))^2 \right], \tag{5}$$

This approach steers the critic toward high-return action sequences in the offline dataset without the potential of exploiting critic approximation errors, preventing value overestimation and enabling stable value learning even with longer action sequences. In Appendix E, we provide the theoretical proof of extending action sequences in detached value learning without loss of generality.

**Distributional RL for enhanced stability** Even with detached value learning, the cumulative reward term $\hat{R}_{t:t+H-1}$ could introduce significant variance when $H$ is large. To enhance stability, we extend our framework with distributional RL (Bellemare et al., 2017; Farebrother et al., 2024), modeling both critic and value networks as categorical distributions over fixed support $[\mathbf{v}_{\min}, \mathbf{v}_{\max}]$ discretized into $m$ bins:

$$Q(s, \mathbf{a}; \theta) = \mathbb{E}[Z(s, \mathbf{a}; \theta)] \quad Z(s, \mathbf{a}; \theta) = \sum_{i=1}^m \hat{p}_i(s, \mathbf{a}; \theta) \cdot \delta_{z_i} \quad \hat{p}_i(s, \mathbf{a}; \theta) = \frac{e^{l_i(s, \mathbf{a}; \theta)}}{\sum_{i=1}^m e^{l_i(s, \mathbf{a}; \theta)}}, \tag{6}$$

where $\hat{p}_i$ denotes the predicted probabilities for location $z_i$. $V(s; \psi)$ is computed similarly, by conditioning only on the state $s$. To address scale differences between regression and classification objectives, we maintain IQL's weighting scheme but replace regression with classification-based learning:

$$\mathcal{L}_V(\psi) = \mathbb{E}_{(s_t, \mathbf{a}_t) \sim \mathcal{D}} \left[ -\alpha_t \cdot \sum_{i=1}^m \hat{p}_i(s_t, \mathbf{a}_t; \bar{\theta}) \log \hat{p}_i(s_t; \psi) \right]$$

$$\alpha_t = \begin{cases} \tau & \text{if } \bar{Q}(s_t, \mathbf{a}_t; \bar{\theta}) \geq V(s_t; \psi) \\ 1 - \tau & \text{otherwise}, \end{cases} \tag{7}$$

$$\mathcal{L}_Q(\theta) = \mathbb{E}_{(s_t, \mathbf{a}_t, s_{t+H}) \sim \mathcal{D}} \left[ -\sum_{i=1}^m p_i \log \hat{p}_i(s_t, \mathbf{a}_t; \theta) \right]. \tag{8}$$

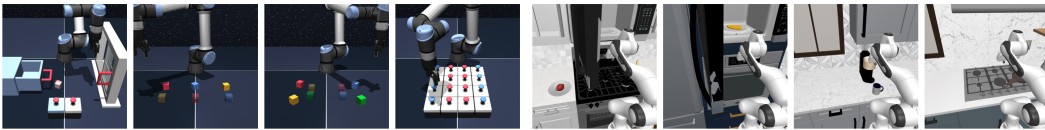

Figure 2: **Simulation task examples.** We study DEAS on 30 different tasks from OGBench (Park et al., 2025a) and 4 challenging manipulation tasks from RoboCasa Kitchen (Nasiriany et al., 2024).

For target probabilities $p_i$, we adopt the truncated normal distribution with mean as Bellman target $(\hat{\mathcal{T}}\mathcal{V})(s_t; \psi) = \sum_{k=0}^{H-1} \gamma_1^k r_{t+k} + \gamma_2^H V(s_{t+H}; \psi)$ and standard deviation $\sigma = 0.75 \cdot (v_{max} - v_{min}/m)$, inspired by Farebrother et al. (2024).

**Dual discount factors**  To further enhance stability and expressiveness in value estimation, we employ two separate discount factors: $\gamma_1$ for rewards within $H$-step action sequences and $\gamma_2$ for the summation across sequence-level decision points. This dual-discounting scheme enables the value function to appropriately weigh immediate and future returns, mitigating issues such as value explosion or collapse that can arise from improper return scaling. In our experiments, we observe that decreasing $\gamma_1$ and increasing $\gamma_2$ lead to more stable training and are critical for performance improvement, especially as the action sequence lengthens (see Section 5.4 for supporting results).

**Compatible policy methods**  For obtaining final policy $\pi(s; \phi)$, our framework is compatible with a variety of policy extraction strategies (Park et al., 2024), including weighted behavior cloning (Peng et al., 2019), deterministic policy gradient (DPG) (Fujimoto & Gu, 2021), best-of-N sampling (Chen et al., 2023), and flow-matching approaches (Park et al., 2025c). Since value function training does not require querying the policy, it can be performed independently, and the policy can be updated separately. To demonstrate this, we illustrate the effectiveness of our method using various policy extraction methods in our experiments.

## 5 EXPERIMENTS

We first validate the effectiveness of DEAS through extensive experiments on various complex, long-horizon tasks in OGBench (Park et al., 2025a). Additionally, to prove that DEAS can be naturally plugged into large-scale VLAs for practical applications, we evaluate DEAS by fine-tuning GR00T N1.5 (NVIDIA, 2025) using offline RL methods on 4 hard tasks from RoboCasa Kitchen (Nasiriany et al., 2024) and also conduct real-world experiments with Franka Emika Research 3 Robot Arm. See Figure 2 and Figure 4 for task examples used in our experiments.

### 5.1 OGBENCH EXPERIMENTS

**Setup**  We evaluate on 6 manipulation environments from OGBench (Park et al., 2025a), each with 5 subtasks. We use datasets ranging from 1M to 100M transitions based on task difficulty. While OGBench is originally designed for offline goal-conditioned RL, we use its single-task variants ('singletask') for reward-maximizing offline RL. For fair comparison, all methods use identical MLP architectures for actor networks and adopt the same policy extraction approach as FQL (Park et al., 2025c), except for CQN-AS, which uses value function networks as the actor itself through discretization. Action sequence length $H$ is set to 8 for scene and puzzle tasks, and 4 for cube tasks, with $n = H$ is used for $n$-step FQL. More details about the experimental setup can be found in Appendix B.1.

**Baselines**  We compare against FQL (Park et al., 2025c), a state-of-the-art offline RL method using one-step distillation between flow matching models with different denoising steps, and $n$-step FQL (Sutton & Barto, 2018), which extends FQL with $n$-step TD updates for horizon reduction (Park et al., 2025b). While increasing $n$ increases bias in standard offline RL, DEAS explicitly models action sequences while maintaining horizon reduction benefits. We also consider Q-Chunking (QC) (Li et al., 2025b), which uses action chunking for actor-critic training while keeping the interaction between actor and critic, while DEAS uses detached value learning. For fair comparison with ours, we extensively tune QC-FQL hyperparameters to achieve higher performance than the original paper. Lastly, CQN-AS (Seo & Abbeel, 2025), a value-based RL method with action sequence utilizing multi-level critics with iterative discretization, is included as a baseline.

Table 1: **Offline RL results** in 6 task categories from OGBench (Park et al., 2025a). We report the success rate (%) and 95% stratified bootstrap confidence interval over 4 runs. **Bold** indicates the values at or above 95% of the best performance. Please refer to Table 9 for the full results.

| Task Category | #Data | FQL | N-step FQL | QC-FQL | CQN-AS | **DEAS** |
|---|---|---|---|---|---|---|
| `scene-play-singletask (5 tasks)` | | $50_{\pm3}$ | $36_{\pm2}$ | $\mathbf{73}_{\pm2}$ | $1_{\pm1}$ | $\mathbf{76}_{\pm2}$ |
| `cube-double-play-singletask (5 tasks)` | 1M | $14_{\pm2}$ | $4_{\pm2}$ | $41_{\pm3}$ | $2_{\pm1}$ | $\mathbf{48}_{\pm2}$ |
| `puzzle-3x3-play-singletask (5 tasks)` | | $44_{\pm3}$ | $36_{\pm3}$ | $62_{\pm7}$ | $0_{\pm0}$ | $\mathbf{91}_{\pm3}$ |
| `cube-triple-play-singletask (5 tasks)` | 10M | $10_{\pm3}$ | $23_{\pm2}$ | $\mathbf{83}_{\pm4}$ | $0_{\pm0}$ | $\mathbf{82}_{\pm5}$ |
| `puzzle-4x4-play-singletask (5 tasks)` | | $32_{\pm4}$ | $19_{\pm5}$ | $69_{\pm8}$ | $0_{\pm0}$ | $\mathbf{82}_{\pm6}$ |
| `cube-quadruple-play-singletask (5 tasks)` | 100M | $17_{\pm8}$ | $36_{\pm10}$ | $45_{\pm7}$ | $0_{\pm0}$ | $\mathbf{64}_{\pm8}$ |

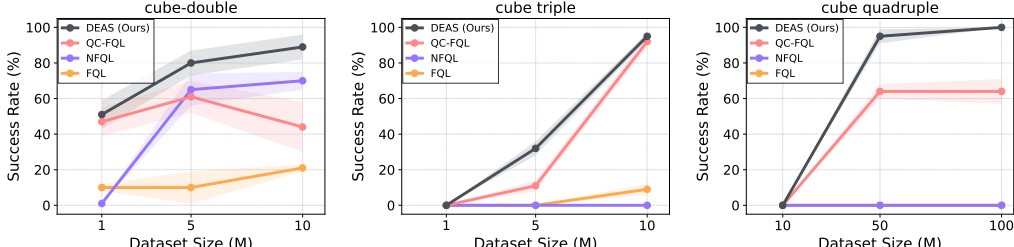

Figure 3: Agent performance across varying dataset sizes on three representative OGBench (Park et al., 2025a) tasks, evaluated by success rate (%). Solid lines indicate the mean, while shaded areas denote the stratified bootstrap confidence intervals over 4 independent runs.

**Quantitative results** As shown in Table 1, DEAS consistently achieves the best performance across all 6 task categories with various dataset sizes. Comparing FQL and N-step FQL, we observe that simply increasing the $n$-step mostly leads to performance degradation due to bias in standard offline RL, while our detached value learning approach enables stable training with action sequences. Notably, DEAS matches or outperforms QC-FQL across all tasks, demonstrating the effectiveness of our stable value learning in addressing offline RL instability. The method shows particularly strong performance on tasks requiring long-horizon reasoning like `puzzle` and the most challenging tasks (i.e., `cube − quadruple`), where the benefits of using action sequences are most pronounced. CQN-AS shows significantly lower performance, likely due to its direct application of strong BC regularization on the value function in the presence of predominantly suboptimal data, along with cumulative errors from iterative discretizations that reduce action precision.

**Scaling analysis** To further validate the scalability of DEAS, we conduct a scaling analysis on three representative OGBench tasks with varying dataset sizes. As shown in Figure 3, DEAS consistently outperforms all baselines across all dataset sizes, achieving the highest success rates in every environment. The method demonstrates robust scaling across different dataset sizes, maintaining consistent performance gains even with larger datasets. This superior performance validates our approach of explicitly modeling action sequences while effectively leveraging suboptimal data through our detached value learning and stable multi-step training.

## 5.2 VLA EXPERIMENTS

To validate the practical applicability of DEAS, we demonstrate its effectiveness with large-scale VLAs (Black et al., 2025; Bjorck et al., 2025; NVIDIA, 2025). These models, trained on internet-scale diverse datasets with billion-scale parameters, predict much longer action sequences and are widely used in robotics applications. However, deploying these models typically requires fine-tuning on task-specific data, which often necessitates collecting expensive expert demonstrations. We design our experiments to validate whether DEAS can improve VLA performance by effectively utilizing suboptimal demonstrations alongside limited expert data, potentially reducing the required amount of costly expert demonstrations. See Appendix B.2 for more details.

### 5.2.1 ROBOCASA KITCHEN EXPERIMENTS

**Setup** We employ GR00T N1.5 (NVIDIA, 2025) and $\pi_0$ (Black et al., 2025) as the backbone VLA. First, we fine-tune the VLA using 100 expert demonstrations from all 24 RoboCasa Kitchen

Table 2: **RoboCasa Kitchen evaluation results**. We fine-tune VLAs on 24 RoboCasa Kitchen tasks using 100 expert demonstrations per task. For 4 selected tasks, we collect 300 rollouts and apply offline IL/RL algorithms. Success rates (%) on 50 episodes, aggregated with 3 seeds. `PnPC2M` denotes 'PnPCounterToMicrowave' and `PnPM2C` denotes 'PnPMicrowaveToCounter'. **Bold** and underline indicate best and runner-up results, respectively.

| Models | CoffeeSetupMug | PnPC2M | PnPM2C | TurnOffStove | Avg. |
|---|---|---|---|---|---|
| | | [*] Result from Bjorck et al. (2025) | | [†] Reproduced performance | |
| *Base model* | | | | | |
| GR00T N1[*] | 2.0 | 0.0 | 0.0 | 15.7 | 4.4 |
| GR00T N1.5[†] | 4.7 | 21.3 | 7.3 | 14.7 | 12.0 |
| $\pi_0$[†] | 20.0 | 11.3 | 10.0 | 8.0 | 12.3 |
| *Imitation learning* | | | | | |
| GR00T N1.5 + Filtered BC | 14.7 | 25.3 | 14.7 | **19.3** | 18.5 |
| $\pi_0$ + Filtered BC | 30.7 | 16.7 | 14.7 | 10.0 | 18.0 |
| *Offline RL* | | | | | |
| GR00T N1.5 + IQL | 23.3 | 30.0 | 14.7 | 12.7 | 20.2 |
| GR00T N1.5 + QC | 16.0 | 28.7 | 14.7 | 10.7 | 17.5 |
| **GR00T N1.5 + DEAS (Ours)** | 28.7 | **36.0** | 18.0 | 18.0 | **25.2** |
| $\pi_0$ **+ DEAS (Ours)** | **37.3** | 15.3 | **19.3** | 15.3 | 21.8 |

tasks to verify that we achieve performance similar to the original GR00T N1 (Bjorck et al., 2025). From these tasks, we select 4 tasks with the lowest success rates in their respective categories for our offline IL/RL experiments. We then collect 300 rollouts for each task from the resulting policy and apply various offline IL/RL methods. For RL methods, we fine-tune the base policy using behavior cloning on both expert demonstrations and the rollout dataset and use the model as an actor for training critic functions when necessary. For policy extraction, we adopt best-of-N sampling (Chen et al., 2023; Nakamoto et al., 2024), where we sample multiple outputs from the policy and select the action sequence with the highest Q-value. We set $H = 16$ in GR00T N1.5 and $H = 50$ in $\pi_0$ for all methods to match the original action chunk size used for each model.

**Baselines** We compare against several baselines across both imitation learning and reinforcement learning paradigms. For imitation learning, we consider Filtered BC, which fine-tunes the base policy using both expert demonstrations and successful episodes from the rollout data (Oh et al., 2018). For reinforcement learning, we evaluate IQL, a value-based method that operates on single actions without requiring policy outputs. For determining action sequence in IQL, we use the very first action in the sequence for value estimation. Lastly, we consider QC, which employs action chunking for critic training but relies on predicted action sequences from VLA for the critic update.

**Results** As shown in Table 2, DEAS achieves the highest success rates in 3 out of 4 tasks, with the remaining task also showing improved performance compared to the base model. While filtered BC improves performance with simple approaches, our approach further improves performance by effectively leveraging suboptimal data. While single-step IQL also demonstrates effectiveness, it yields smaller performance gains across all tasks than our approach, due to its lack of action-sequence understanding. QC shows limited improvement compared to BC-based approaches, highlighting the advantage of our detached value learning with action sequences. A similar pattern holds for the $\pi_0$ backbone: DEAS yields clear improvements over the base model across all tasks, even with a significantly longer action sequence (50) than the GR00T N1.5 backbone. This confirms that our method reliably enhances VLA policies regardless of backbone and action-sequence length.

### 5.2.2 REAL-WORLD EXPERIMENTS

**Setup** We further investigate the effectiveness of DEAS in real-world tasks using Franka Emika Research 3 Robot Arm. Inspired by RoboCasa Kitchen, we design pick-and-place tasks from the countertop to the bottom cabinet, with three different objects: `peach`, `milka`, and `hichew` (see Figure 4). For each task, we collect 5 demonstrations, fine-tune GR00T N1.5, collect 25 rollouts, and apply various offline IL/RL methods. We evaluate using 20 rollouts per task from 5 different initial points and use the same baselines as in the RoboCasa Kitchen experiments. Success rates are calculated based on partial success scoring (0-1 scale) that considers subtask completion, with detailed evaluation methodology provided in Section B.2.2.

Figure 4: **Real-world tasks.** We conduct pick-and-place tasks from the countertop to the bottom cabinet with `peach`, `milka`, and `hichew`.

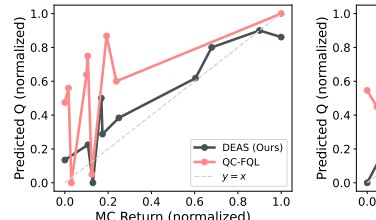

Table 3: **Real-world evaluation results.** We report the partial success rate (%, over 20 trials per task) on 3 tasks from 5 initial points. **Bold** and underline indicate best and runner-up results, respectively.

| Models | `peach` | `milka` | `hichew` | Avg. |
|---|---|---|---|---|
| *Base model* | | | | |
| GR00T N1.5 | 62.0 | 45.0 | 85.0 | 64.0 |
| *Imitation learning* | | | | |
| + Filtered BC | 76.3 | 25.0 | 92.5 | 64.6 |
| *Offline RL* | | | | |
| + IQL | 82.5 | 37.5 | 78.8 | 66.3 |
| + QC | 58.8 | 15.0 | 45.0 | 39.6 |
| **+ DEAS (Ours)** | **86.3** | **53.8** | **95.0** | **78.4** |

**Results**  In Table 3, DEAS achieves the highest success rates across all three pick-and-place tasks compared to baselines. The method shows consistent improvements, particularly on challenging objects like `milka` (a deformable object) where other approaches struggle. Notably, QC shows degraded performance compared to the base model, likely due to its instability when using action sequences with relatively small datasets, while our method shows stable improvement even with limited data. These results demonstrate that our detached value learning approach can be effectively applied to real-world robotic tasks and remains stable regardless of the dataset size.

## 5.3 QUANTITATIVE ANALYSES

**Alignment to actual returns**  To evaluate whether a critic with action sequences is properly aligned with actual returns, we analyze value calibration between the critic values and the true discounted returns. The main intuition of this experiment is that a properly trained critic should maintain a monotonic relationship between predicted Q values and actual returns on unseen state-action distributions. First, we compute critic pre-

Figure 5: Value calibration curve comparing predicted critic values to Monte-Carlo returns on held-out trajectories in puzzle-4x4 (left) and cube-quadruple (right).

dictions $\hat{Q}(s_t, \mathbf{a}_t)$ and Monte-Carlo returns $\hat{G}(s_t, \mathbf{a})$ from transitions of 5000 unseen trajectories. We then partition transitions into bins by $\hat{Q}$ values and plot the average $\hat{G}$ within each bin. The deviation from the diagonal $y = x$ quantifies overestimation (above) or underestimation (below). As shown in Figure 5, DEAS exhibits a calibration curve consistently closer to the diagonal than QC-FQL, indicating reduced overestimation and improved value alignment. These findings support the claim that DEAS's detached value learning framework with distributional objectives provides more reliable value estimation than baselines that incorporate actor-critic interactions in extended, high-dimensional action spaces.

**Robustness to data qualities**  To validate the robustness of DEAS across dataset qualities, we construct mixed datasets by combining `play` and `noisy` datasets provided by OGBench, where `noisy` datasets contain more suboptimal transitions, at different ratios. We measure success rates and analyze calibration curves of DEAS and QC-FQL in Figure 6. DEAS outperforms QC-FQL across all data regimes, demonstrating robust performance under varying data quality. Additionally, DEAS's calibration curve remains consistently closer to the diagonal across all regimes, which further supports the effectiveness of our method.

## 5.4 ABLATION STUDIES

We investigate the effect of hyperparameters and various components of DEAS by running experiments on OGBench puzzle-4x4 task.

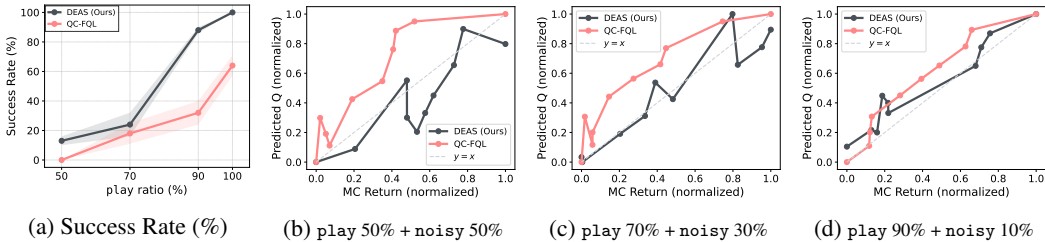

Figure 6: Success rate (%) (a) and value calibration curve (b,c,d) across different mixtures on cube-quadruple task. DEAS exhibits not only higher task performance but also an improved value calibration closer to the diagonal across all data regimes.

| $H$ | Actor | SR |
|-----|--------|------|
| 1 | $512 \times 4$ | $21 \pm 3$ |
| 2 | $512 \times 4$ | $25 \pm 5$ |
| 4 | $512 \times 4$ | $75 \pm 8$ |
| 8 | $512 \times 4$ | $\mathbf{88} \pm 4$ |
| 16 | $512 \times 4$ | $51 \pm 4$ |
| 16 | $1024 \times 4$ | $84 \pm 4$ |

(a) Action sequence

| Critic | Value | SR |
|--------|-------|------|
| $256 \times 4$ | $256 \times 4$ | $69 \pm 7$ |
| $512 \times 4$ | $256 \times 4$ | $\mathbf{88} \pm 4$ |
| $1024 \times 4$ | $256 \times 4$ | $\mathbf{91} \pm 4$ |
| $512 \times 4$ | $512 \times 4$ | $50 \pm 4$ |

(b) Critic size

| IQL | HLG | SR |
|-----|-----|------|
| ✗ | ✓ | $75 \pm 5$ |
| ✓ | ✗ | $63 \pm 6$ |
| ✓ | ✓ | $\mathbf{88} \pm 4$ |

(c) Objectives

| $\gamma_1$ | $\gamma_2$ | SR |
|-----------|-----------|------|
| 0.8 | 0.999 | $\mathbf{87} \pm 4$ |
| 0.9 | 0.999 | $\mathbf{88} \pm 4$ |
| 0.99 | 0.999 | $81 \pm 5$ |
| 0.999 | 0.999 | $80 \pm 8$ |

(d) $\gamma_1$ and $\gamma_2$

Table 4: **Ablation studies.** We investigate the effect of (a) action sequence length $H$, (b) critic and value model size, (c) training objectives, and (d) separate discount factors $\gamma_1$ and $\gamma_2$. SR denotes success rate (%) and default settings are highlighted in  gray . **Bold** indicates values at or above 95% of the best performance.

**Effect of action sequence length**   Table 4a investigates the impact of action sequence length on performance. When using single-step or two-step action ($H = 1, 2$), DEAS fails to achieve meaningful performance, confirming the necessity of action sequences for long-horizon tasks. Performance improves with longer sequences, but when the sequence length becomes longer than 8, it requires proportionally larger actor networks to handle the increased action dimensions, suggesting a trade-off between sequence length and computational efficiency.

**Effect of network size**   Table 4b analyzes the sensitivity to network sizes. For the critic network, we observe that increasing capacity initially improves performance by better approximating the value function. For the value function, we find that the network needs sufficient capacity to capture the complexity of action sequence values, but excessive capacity without proper regularization causes instability in value estimation, leading to performance degradation.

**Effect of training objective**   In Table 4c, we compare different training objectives for value estimation. We found that using only distributional RL (HLG) (Farebrother et al., 2024) or only standard regression (IQL) shows limited performance. However, combining detached value learning with distributional estimation significantly improves results, suggesting both components are crucial for stable training with action sequences.

**Effect of dual discount factors**   Lastly, we examine the effect of dual discount factors on learning dynamics in Table 4d. Proper tuning of $\gamma_1$ (the discount factor for action sequences) is essential for performance, as it controls the temporal horizon for value estimation within sequences. In this paper, we use $\gamma_1 = 0.9$ for all experiments.

## 6   CONCLUSION

We introduced DEAS, a simple but effective offline RL method that exploits action sequences to learn efficiently on complex tasks. DEAS provides a practical recipe for modeling temporally extended actions while avoiding value overestimation via detached value learning and distributional networks, yielding a principled reduction of the effective planning horizon that is critical in long-horizon settings. Empirically, it consistently outperforms strong baselines on challenging OGBench tasks and scales to large VLAs across simulation and real-world experiments, demonstrating its potential to bring offline RL closer to real-world applications.

REPRODUCIBILITY STATEMENT

We provide full hyperparameter and implementation details in Section 5 and Section B. In addition, to facilitate reproduction, we release an open-source implementation on the project website.

ACKNOWLEDGMENTS

CY thanks Jaehyun Nam, Juyong Lee, Yonghoon Dong, and anonymous reviewers for providing helpful feedback and suggestions for improving our paper. CY also thanks Angeline S. Kim for assistance with improving the paper's expression and visualization. This work was partly supported by Institute for Information & communications Technology Planning & Evaluation (IITP) grant funded by the Korea government (MSIT) (RS-2019-II190075, Artificial Intelligence Graduate School Program(KAIST); RS-202400509279, Global AI Frontier Lab) and the National Research Foundation of Korea (NRF) grant funded by the Korea government(MSIT) (RS-2024-00414822).

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

## A LIMITATIONS AND FUTURE WORK

While DEAS demonstrates significant improvements over existing offline RL methods, several limitations and opportunities for future research remain. First, our current approach uses fixed action sequence lengths across different tasks, whereas optimal sequence lengths vary significantly depending on task complexity. One natural direction is to extend DEAS to even longer horizons, potentially with explicit hierarchical or option-based sequence generation. The other is to investigate adaptive mechanisms that can dynamically adjust action sequence lengths based on task requirements, potentially by adopting hierarchical policies (Kulkarni et al., 2016; Vezhnevets et al., 2017; Nachum et al., 2018), which would be an intriguing research direction.

Second, while DEAS shows promising results on individual tasks, scaling to large-scale unified value functions remains a critical challenge for real-world deployment. DEAS currently trains reward models on 3-4 tasks simultaneously, but practical applications require learning from hundreds or thousands of diverse tasks. Future research should focus on developing scalable architectures and training procedures that can handle massive multi-task datasets while maintaining sample efficiency and avoiding catastrophic forgetting.

Third, our method relies on distributional RL with fixed support ranges ($\mathbf{v}_{\min}, \mathbf{v}_{\max}$) and discretization parameters, which can significantly impact performance. The sensitivity to these hyperparameters limits the method's robustness across different domains and reward scales. Future work should develop more robust frameworks that can automatically adapt to different reward distributions or provide principled ways to set these parameters.

## B IMPLEMENTATION AND TRAINING DETAILS

### B.1 OGBENCH EXPERIMENTS

**Tasks** We evaluate our method on 6 UR5 Robot Arm manipulation environments from OG-Bench (Park et al., 2025a), each with 5 subtasks. All tasks are state-based, and goal-free setup. For each task, the observation space consists of the proprioceptive state of the UR5 Robot Arm, and low-dim state vector informing the target object state and position. The action space consists of the cartesian position of UR5 robot arm, gripper yaw, and gripper open/close. For substituting goal-conditioned environment to standard function, we use the simple semi-sparse reward function, which is defined as the negative number of uncompleted subtasks in the current state, following Park et al. (2025a). For all tasks, the maximum episode length is set to 1000.

**Implementation details** We implement our method on top of the open-source implementation of FQL (Park et al., 2025c) [1]. Unless otherwise mentioned, we largely follow the training/evaluation setup and network architecture from Park et al. (2025c) and Park et al. (2025b). For training value network, we use the smaller size network compared to critic network for all experiments, which shows the best performance, and we use the doubled size of network for the critic network. For `cube` experiments, we use BRO (Nauman et al., 2024) for additional regularization between relatively small range of returns in value function training. For selecting $\mathbf{v}_{\min}$ and $\mathbf{v}_{\max}$ for distributional RL, we use two procedures: 1) *data-centric*: compute return distribution from the dataset and select 1% and 99% quantiles with 20% padding, and 2) *universal*: compute theoretical bounds using reward range $[r_{\min}, r_{\max}]$, horizon $L$, action sequence length $H$, and discount factors $\gamma_1, \gamma_2$. In this case, the theoretical bounds are:

$$\mathbf{v}_{\min} = r_{\min} \frac{1 - \gamma_2^H}{1 - \gamma_2} \frac{1 - \gamma_1^K}{1 - \gamma_1} \tag{9}$$

$$\mathbf{v}_{\max} = r_{\max} \frac{1 - \gamma_2^H}{1 - \gamma_2} \frac{1 - \gamma_1^K}{1 - \gamma_1} \tag{10}$$

where $\gamma_1$ and $\gamma_2$ denote discount factors for inner-sequence and across-sequence-level decision, respectively.

---

[1] https://github.com/seohongpark/fql

**Training and evaluation**   For the training dataset, we use the open-sourced 1M/100M `play` dataset released by Park et al. (2025a) [2], where the dataset is collected by open-loop, non-Markovian scripted policies with temporally correlated noise. As 100M dataset consists of 100 separate files with 1M transitions for each, we use the first 10 files sorted by name for 10M dataset. We train our method and baselines for 1M (1M data) / 2.5M (10M/100M data) gradient steps. For selecting BC coefficient $\alpha$ for policy extraction, we first normalize the Q loss as in Fujimoto & Gu (2021) and sweep the value from $\{0.1, 0.3, 1, 3, 10\}$ and choose the best one for each task and baseline, except $cube - double$, where we follow the hyperparameter used in Li et al. (2025b). For evaluation, we report the average success rates across the last three evaluation epochs (800K, 900K, 1M for 1M dataset, 2.3M, 2.4M, 2.5M for 10M/100M dataset) following Park et al. (2025c) and Park et al. (2025b). For checking additional hyperparameters used in our experiments, please refer to Section B.3.

**Baselines**   For reporting results from FQL and $n$-step FQL, we use the implementation from Park et al. (2025c). For Q-Chunking, we re-implement the code from Li et al. (2025b) [3] in our codebase. We found that simply increasing discount factor $\gamma$ leads to significant performance improvement for Q-Chunking, so we use the discount factor to be same with $\gamma_2$ for value function training. For implementing CQN-AS, we use the original implementation released by the authors from Seo & Abbeel (2025) [4] and integrate OGBench related codes on top of the codebase. Originally, CQN-AS is designed to apply auxiliary BC loss only on expert demonstrations, but considering the dataset distribution of OGBench tasks with nearly no success rollouts, we modify the BC loss on the suboptimal data as well (Fujimoto & Gu, 2021; Park et al., 2025c; 2024), where no significant difference with the original implementation. As the reward scale for OGBench is highly different according to the domain, we normalize the reward scale to be in $[-1, 0]$, and use $\mathbf{v}_{\min}$ and $\mathbf{v}_{\max}$ as $-200$ and $0$, respectively. For levels and bins, we use 5 (level) and 9 (bins) for all experiments.

**Computing hardware**   For all OGBench experiments, we use a single NVIDIA RTX 3090 GPU with 24GB VRAM and it takes about 2 hours for training the small model (used for 1M dataset) and about 8 hours for training the large model (used for 10M/100M dataset).

## B.2   VLA EXPERIMENTS

**Computing hardware**   For all VLA experiments, we use NVIDIA A100 80GB GPUs. Fine-tuning GR00T N1.5 takes about 4 hours for 100 expert demonstrations and successful rollouts. For training DEAS and baselines, it takes about 10 hours with the same data, as we use a larger batch size.

**VLA fine-tuning**   We implement our method and baselines on top of the open-source implementation of GR00T N1.5 (NVIDIA, 2025) [5]. As our code is based on an earlier version of GR00T N1.5, we conduct experiments without introducing future tokens to the action expert modules. For fine-tuning GR00T N1.5, we use a batch size of 32 and train for 30K (RoboCasa Kitchen) / 10K (Real Robot) steps using AdamW (Loshchilov & Hutter, 2019) optimizer with learning rate $1 \times 10^{-4}$ and cosine annealing schedule.

### B.2.1   ROBOCASA KITCHEN EXPERIMENTS

**Task**   RoboCasa Kitchen (Nasiriany et al., 2024) is a simulation environment with a mobile manipulator attached to a Franka Panda robot arm in household kitchen environments. Among 24 atomic tasks provided by the environment, we select 4 challenging tasks (`CoffeeSetupMug`, `PnPMicrowaveToCounter`, `PnPMicrowaveToMicrowave`, `PnPMicrowaveToStove`) that require relatively long-horizon and delicate manipulation with small grasping part, which is demonstrated by the low success rate of the base model. For perception, camera images from 3 different viewpoints (left front, right front, wrist), proprioceptive states including position/velocities of joint/base, and natural language instructions, are provided. For reward function, we use the pre-defined success detector in the environment, and use the sparse reward function where the reward is 1 if the task is

---

[2] https://github.com/seohongpark/ogbench
[3] https://github.com/ColinQiyangLi/qc
[4] https://github.com/younggyoseo/CQN-AS
[5] https://github.com/NVIDIA/Isaac-GR00T

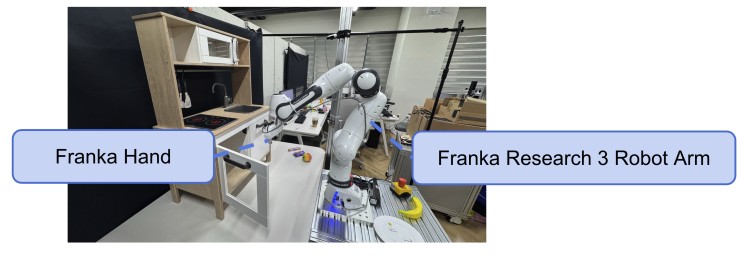

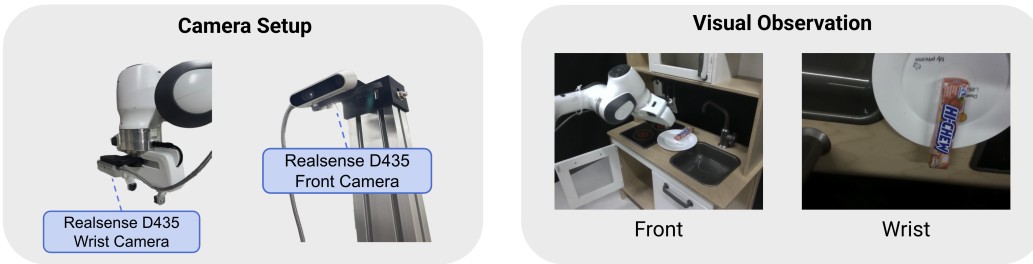

Figure 7: **Real-robot platform.**

completed, and 0 otherwise. Following previous works (Kumar et al., 2023b; Mark et al., 2024), we utilized the practical heuristic of annotating the last $n = 15$ transitions of every successful trajectory with a reward of +1.

**Implementation details** As an input for the value function, we first use the proprioceptive states from the robot, including joint position/angle, base position/orientation for the mobile manipulator. To provide information on target objects to the value function, we utilize the encoded representation of three different camera views and task instructions from the VLM backbone. For the value/critic network architecture, we use the same hyperparameters as those used for the 100M dataset experiments. For optimizing value and critic function, we use the expectile parameter $\tau$ as 0.7, $\gamma_1 = 0.9$, $\gamma_2 = 0.99$, and *universal* support type for distributional RL, for all experiments. For selecting action candidates with the value function, we first sample $N = 10$ candidates from the policy. For selecting final actions, we try either 1) greedy sampling with highest Q-value or 2) inspired by Nakamoto et al. (2024), sampling the action from a categorical distribution obtained by temperature controlled softmax over Q-values: $a_t \sim \text{Softmax}(\frac{Q(s_t, a_1)}{\beta}, \ldots, \frac{Q(s_t, a_N)}{\beta})$ with temperature $\beta = 1$ and report the best result for each task.

**Training and evaluation** For expert demonstrations, we randomly sample 100 expert demonstrations using the publicly available dataset generated by MimicGen (Mandlekar et al., 2023). For training DEAS and baselines, we use a batch size of 64 and train for 30K steps using Adam optimizer with a learning rate of $3 \times 10^{-4}$. For collecting rollouts, we use randomized environments using the object instance set $A$. For each task, we evaluate the model performance across 50 trials on five distinct evaluation scenes with 3 different evaluation seeds, totaling 150 rollouts. To test generalization capabilities, we evaluate the policy only on unseen object instances.

### B.2.2 REAL ROBOT EXPERIMENTS

**Hardware platform** We use Franka Research 3, a 7-DoF robotic arm, for our experiments. For visual perception, we utilize the dual camera with Intel RealSense D435i: a camera attached to the column next to the robot base to provide a global view, and a wrist-mounted camera for a close-range view. Teleoperated demonstrations are collected using an Oculus Quest 2, and we log time-synchronized RGB images, joint states, and gripper width for data collection. Demonstrations are recorded at 15 Hz. See Figure 7 for visual examples.

**Task** We evaluate the model performance on pick-and-place tasks from the countertop to the bottom cabinet, with three different objects: `peach`, `milka`, and `hichew`. Each object has different

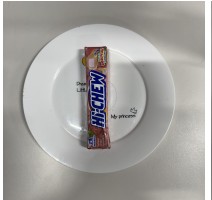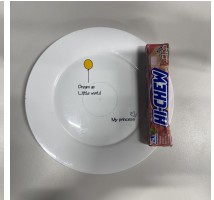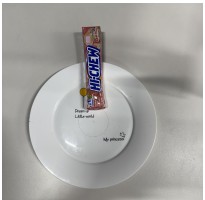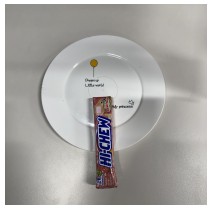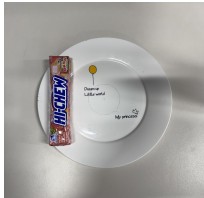

Figure 8: **Initialization points used for pick-and-place tasks.**

properties: `peach` is a rigid object with a relatively larger size that is easy to occlude, `milka` is a deformable object with a relatively smaller size that is easy to deform, and `hichew` is a hard object requiring precise grasping due to its small width. For collecting demonstrations, we use different initialization points (center, top, bottom, left, right) and collect one demonstration for each position (see Figure 8 for the initialization points used in our experiments). For accurate value function estimation, we manually label the reward function for each task. Specifically, we split the task into 4 stages: 1) moving to the countertop, 2) picking up the object, 3) moving to the target position, and 4) placing the object. For each stage, we label the reward function as 1 if the task is completed, and 0 otherwise, and we set the reward function as the negative number of uncompleted stages following Park et al. (2025a).

**Implementation details**    Unless otherwise mentioned, we follow the same implementation details as in the RoboCasa Kitchen experiments. For selecting final actions, we use $N = 50$ candidates from the policy and use the same procedure for selecting the final action as in the RoboCasa Kitchen experiments.

### B.3    HYPERPARAMETERS

We list the hyperparameters used in our OGBench experiments in Tables 5 and 6. For the BC coefficient $\alpha$ used for policy extraction, please refer to Table 7.

Table 5: **DEAS hyperparameters for OGBench experiments.**

| Hyperparameter | Value |
|---|---|
| Gradient steps | 1M (1M dataset), 2.5M (10M/100M dataset) |
| Optimizer | Adam (Kingma, 2015) |
| Learning rate | 0.0003 |
| Batch size | 256 (1M dataset), 1024 (10M/100M dataset) |
| Actor MLP size | $[512, 512, 512, 512]$ (1M dataset) |
|  | $[1024, 1024, 1024, 1024]$ (10M/100M dataset) |
| Critic MLP size | $[256, 256, 256, 256]$ (1M dataset) |
|  | $[512, 512, 512, 512]$ (10M/100M dataset) |
| Value MLP size | $[128, 128, 128, 128]$ (1M dataset) |
|  | $[256, 256, 256, 256]$ (10M/100M dataset) |
| Nonlinearity | GELU (Hendrycks & Gimpel, 2016) |
| Layer normalization | True |
| Target network update rate | 0.005 |
| Discount factor $\gamma_1$ | 0.9 |
| Discount factor $\gamma_2$ | 0.995 (`cube`), 0.999 (`scene`, `puzzle`) |
| HL-Gaussian - Atoms | 101 |
| HL-Gaussian - $\sigma$ | 0.75 |
| HL-Gaussian - Support range type | *data-centric* (`cube`), *universal* (`scene`, `puzzle`) |
| Flow steps | 10 |
| Critic ensemble size | 2 |
| Action sequence length $H$ | 4 (`cube`), 8 (`scene`, `puzzle`) |
| Expectile $\kappa$ (DEAS) | 0.9 (1M dataset), 0.95 (10M/100M dataset) |
| Double Q aggregation | $\min(Q_1, Q_2)$ |
| Policy extraction hyperparameters | Table 7 |

Table 6: **Baseline hyperparameters for OGBench experiments.**

| Hyperparameter | Value |
|---|---|
| Critic MLP size | $[512, 512, 512, 512]$ (1M dataset) |
| | $[1024, 1024, 1024, 1024]$ (10M/100M dataset) |
| Discount factor $\gamma$ (FQL, $n$-step FQL) | 0.99 |
| Discount factor $\gamma$ (QC-FQL) | 0.995 `(cube)`, 0.999 `(puzzle)` |
| Horizon reduction factor $n$ | 4 `(cube)`, 8 `(puzzle)` |
| Policy extraction hyperparameters | Table 7 |
| Levels (CQN-AS) | 5 |
| Bins (CQN-AS) | 9 |
| C51 - $\mathbf{v}_{min}, \mathbf{v}_{max}$ (CQN-AS) | -200, 0 |

Table 7: **Policy extraction hyperparameters for OGBench experiments.** Note that we apply Q-Normalization (Fujimoto & Gu, 2021) for actor loss, except `cube-double tasks`.

| Task | FQL $\alpha$ | $n$-step FQL $\alpha$ | QC-FQL $\alpha$ | DEAS $\alpha$ |
|---|---|---|---|---|
| `scene` | 3 | 1 | 3 | 3 |
| `cube-double` | 300 | 100 | 300 | 300.0 |
| `puzzle-3x3` | 3 | 1 | 1 | 3 |
| `cube-triple` | 3 | 1 | 1 | 1 |
| `puzzle-4x4` | 3 | 1 | 1 | 3 |
| `cube-quadruple` | 3 | 1 | 1 | 1 |

## C  EXTENDED RELATED WORK

**Reinforcement learning with VLAs**  Recent efforts have applied RL to VLA training (Zhang et al., 2024; Chen et al., 2025a; Zhang et al., 2025; Guo et al., 2025; Tan et al., 2025; Chen et al., 2025b; Li et al., 2025a), but most focus on on-policy online RL, which requires expensive interactions and cannot reuse transitions. A key limitation is that existing methods use single-step value functions $Q(s, a)$ for value learning, despite modern VLAs being designed to predict action sequences (Black et al., 2025; Bjorck et al., 2025; Intelligence et al., 2025). This mismatch between single-step value learning and multi-step action prediction limits the effectiveness of RL with VLAs. The most related work is CO-RFT (Huang et al., 2025), which applies chunked offline RL to VLA training, but differs from our approach in three key aspects: (1) CO-RFT uses actor-critic methods (Nakamoto et al., 2023) with single-step value functions while DEAS uses detached value learning with action sequences, (2) CO-RFT relies on human teleoperated expert demonstrations while we use small expert sets with large suboptimal rollouts, and (3) CO-RFT requires sophisticated transformer architectures while DEAS achieves improvements with simple MLP networks.

## D  USE OF LARGE LANGUAGE MODELS

We acknowledge the use of large language models (LLMs) in preparing this manuscript. LLMs were employed solely to refine writing quality, including grammar correction, vocabulary suggestions, and typographical checks. All substantive ideas, analyses, and conclusions in this paper are entirely the work of the authors.

# E PROOFS

In this section, we will show that DEAS leveraging action sequences can recover the optimal value function under the dataset support constraints following flows of Kostrikov et al. (2022) with the primitive action $a$ replaced by the sequence action $o \in \mathcal{A}^H$.

**Lemma E.1** (Expectile limit and monotonicity; cf. Lemma 1 of Kostrikov et al. (2022)). *Let $X$ be a real-valued random variable with bounded support and let $x^\star := \sup\{x \in \mathbb{R} : \Pr(X \le x) < 1\}$ denote the supremum of its support. For $\tau \in (0, 1)$, let $m_\tau$ be the $\tau$-expectile of $X$, i.e.,*

$$m_\tau = \arg\min_{m \in \mathbb{R}} \mathbb{E}\big[|\tau - \mathbb{1}\{X < m\}|(X - m)^2\big].$$

*Then:*

1. *For any $0 < \tau_1 < \tau_2 < 1$, we have $m_{\tau_1} \le m_{\tau_2}$.*

2. $\lim_{\tau \to 1} m_\tau = x^\star$.

Let $\mu_H(o \mid s)$ denote the empirical behavior distribution over $H$-step action sequence $\mathbf{a}_t := (a_t, \ldots, a_{t+H-1}) \in \mathcal{A}^H$ extracted from the dataset. Given the multi-step return $\tilde{R}(s_t, \mathbf{a}_t, \gamma_1)$ and discount $\gamma_2$ across sequence-level decision points, we define the optimal solutions of Equation (4) and Equation (5) by

$$V_\tau(s) = \arg\min_{v \in \mathbb{R}} \mathbb{E}_{o \sim \mu_H(\cdot|s)}\big[L_\tau\big(Q_\tau(s, \mathbf{a}) - v\big)\big], \tag{11}$$

$$Q_\tau(s, \mathbf{a}) = \tilde{R}(s, \mathbf{a}, \gamma_1) + \gamma_2^H \mathbb{E}\big[V_\tau(s_{t+H}) \mid s_t = s, \mathbf{a}_t = o\big]. \tag{12}$$

**Lemma E.2** (Action-sequence analogue of Lemma 2). *Let $0 < \tau_1 < \tau_2 < 1$. Then, for all $s \in \mathcal{S}$,*

$$V_{\tau_1}(s) \le V_{\tau_2}(s).$$

*Proof.* The proof of Lemma 2 in Kostrikov et al. (2022) uses (i) Lemma E.1 and (ii) the Bellman recursion for $Q_\tau$ and $V_\tau$. In our setting, Equation (11) and Equation (12) play the role of the original value and Bellman definitions, with $o \in \mathcal{A}^H$ in place of $a \in \mathcal{A}$. Replacing $a$ by $o$ in the argument reproduces the same inequalities verbatim, yielding $V_{\tau_1}(s) \le V_{\tau_2}(s)$ for all $s$. $\qquad\square$

We now define the dataset-constrained optimal value function $Q^\star$ with action sequence $\mathbf{a}_t$ as:

$$Q^\star(s, \mathbf{a}) := \tilde{R}(s, \mathbf{a}, \gamma_1) + \gamma_2^H \mathbb{E}\Big[\max_{o' \in \mathrm{Supp}(\mu_H(\cdot|s_{t+H}))} Q^\star(s_{t+H}, o') \,\Big|\, s_t = s, \mathbf{a}_t = o\Big]. \tag{13}$$

**Corollary E.2.1** (Action-sequence analogue of Corollary 2.1). *For any $\tau \in (0, 1)$ and any $s \in \mathcal{S}$,*

$$V_\tau(s) \le \max_{o \in \mathrm{Supp}(\mu_H(\cdot|s))} Q^\star(s, \mathbf{a}),$$

*where $V_\tau$ is defined in Equation (11) and $Q^\star$ is given by Equation (13).*

*Proof.* For each fixed $s$, the expectile $V_\tau(s)$ is a convex combination of the values $Q_\tau(s, \mathbf{a})$ for $o$ in the support of $\mu_H(\cdot \mid s)$:

$$V_\tau(s) = \sum_i w_i Q_\tau(s, \mathbf{a}_i), \quad w_i \ge 0, \quad \sum_i w_i = 1, \quad o_i \in \mathrm{Supp}(\mu_H(\cdot \mid s)).$$

Therefore

$$V_\tau(s) \le \max_{o \in \mathrm{Supp}(\mu_H(\cdot|s))} Q_\tau(s, \mathbf{a}).$$

$\qquad\square$

**Theorem E.3** (Action-sequence analogue of Theorem 3). *For all $s \in \mathcal{S}$,*

$$\lim_{\tau \to 1} V_\tau(s) = \max_{o \in \mathrm{Supp}(\mu_H(\cdot|s))} Q^\star(s, \mathbf{a}),$$

*where $Q^\star$ is the dataset-constrained optimal sequence-action value function defined in Equation (13).*

*Proof.* Lemma E.1 applies directly to the scalar random variable $Q^{\star}(s, \mathbf{a})$ and implies that the $\tau$-expectile of this random variable converges to its supremum as $\tau \to 1$. Combining this fact with Lemma E.2 and Corollary E.2.1 yields the claimed limit, exactly as in the proof of Theorem 3 in Kostrikov et al. (2022). □

In summary, we show that the results of Section 4.4 in Kostrikov et al. (2022) hold without modification when the single action $a$ is replaced by a fixed-length action sequence $o \in \mathcal{A}^{H}$. Moreover, integrating IQL with action sequences and distributional RL is also theoretically sound, as established by the same arguments in Bellemare et al. (2017; 2023).

# F OGBENCH DATASET STATISTICS

Table 8: Dataset statistics for OGBench tasks.

| Task | Success Rate (%) | Reward | | | | Return | | | |
|---|---|---|---|---|---|---|---|---|---|
| | | Min | Max | Mean | Std | Min | Max | Mean | Std |
| scene-singletask-task1-v0 | 6.0 (60/1000) | -5.0 | 0.0 | -3.173325 | 1.0295831 | -4823.0 | -1.0 | -1557.8765 | 986.0316 |
| scene-singletask-task2-v0 | 4.0 (40/1000) | -5.0 | 0.0 | -3.163389 | 0.9956842 | -4302.0 | 0.0 | -1553.6287 | 971.3409 |
| scene-singletask-task3-v0 | 6.6 (66/1000) | -5.0 | 0.0 | -3.155803 | 1.0406193 | -4765.0 | 0.0 | -1551.8962 | 985.3525 |
| scene-singletask-task4-v0 | 10.1 (101/1000) | -5.0 | 0.0 | -3.091423 | 1.050524 | -4696.0 | 0.0 | -1494.2867 | 965.7741 |
| scene-singletask-task5-v0 | 6.2 (62/1000) | -5.0 | 0.0 | -3.191186 | 1.0178193 | -4530.0 | 0.0 | -1548.5066 | 984.6961 |
| cube-double-singletask-task1-v0 | 0.7 (7/1000) | -2.0 | 0.0 | -1.949015 | 0.22173753 | -2000.0 | 0.0 | -974.9428 | 567.3881 |
| cube-double-singletask-task2-v0 | 1.5 (15/1000) | -2.0 | 0.0 | -1.947426 | 0.22576976 | -2000.0 | 0.0 | -972.7409 | 567.2142 |
| cube-double-singletask-task3-v0 | 1.7 (17/1000) | -2.0 | 0.0 | -1.949817 | 0.22287366 | -2000.0 | -1.0 | -976.7585 | 567.5913 |
| cube-double-singletask-task4-v0 | 1.7 (17/1000) | -2.0 | 0.0 | -1.949545 | 0.2221605 | -2000.0 | 0.0 | -976.39 | 567.5661 |
| cube-double-singletask-task5-v0 | 2.9 (29/1000) | -2.0 | 0.0 | -1.973839 | 0.16801369 | -2000.0 | 0.0 | -988.5528 | 572.3737 |
| puzzle-3x3-singletask-task1-v0 | 6.5 (65/1000) | -9.0 | 0.0 | -4.49132 | 1.4982606 | -5459.0 | 0.0 | -2249.069 | 1317.5267 |
| puzzle-3x3-singletask-task2-v0 | 5.2 (52/1000) | -9.0 | 0.0 | -4.505488 | 1.503788 | -5562.0 | -1.0 | -2252.3616 | 1321.4901 |
| puzzle-3x3-singletask-task3-v0 | 5.5 (55/1000) | -9.0 | 0.0 | -4.498164 | 1.506579 | -5488.0 | 0.0 | -2249.2212 | 1318.7554 |
| puzzle-3x3-singletask-task4-v0 | 5.4 (54/1000) | -9.0 | 0.0 | -4.48985 | 1.4987043 | -5516.0 | 0.0 | -2244.5073 | 1317.8716 |
| puzzle-3x3-singletask-task5-v0 | 4.8 (48/1000) | -9.0 | 0.0 | -4.50203 | 1.4991112 | -5630.0 | -1.0 | -2251.7349 | 1322.249 |
| cube-triple-singletask-task1-v0 | 0.01 (1/10000) | -3.0 | 0.0 | -2.9232252 | 0.2722444 | -3000.0 | -1.0 | -1463.3553 | 850.65326 |
| cube-triple-singletask-task2-v0 | 0.03 (3/10000) | -3.0 | 0.0 | -2.9237301 | 0.27177206 | -3000.0 | -1.0 | -1463.7289 | 850.6014 |
| cube-triple-singletask-task3-v0 | 0.02 (2/10000) | -3.0 | 0.0 | -2.9217937 | 0.275608 | -3000.0 | -1.0 | -1462.3278 | 850.74963 |
| cube-triple-singletask-task4-v0 | 0.04 (4/10000) | -3.0 | 0.0 | -2.9230156 | 0.27358177 | -3000.0 | -1.0 | -1463.4613 | 850.6792 |
| cube-triple-singletask-task5-v0 | 0.17 (17/10000) | -3.0 | 0.0 | -2.9675632 | 0.1871572 | -3000.0 | 0.0 | -1484.8967 | 859.6889 |
| puzzle-4x4-play-singletask-task1-v0 | 0.02 (2/10000) | -16.0 | 0.0 | -8.000628 | 1.9951512 | -10471.0 | -1.0 | -4006.1116 | 2348.2068 |
| puzzle-4x4-play-singletask-task2-v0 | 0.06 (6/10000) | -16.0 | 0.0 | -7.999522 | 1.9958299 | -10532.0 | -1.0 | -4005.195 | 2349.1113 |
| puzzle-4x4-play-singletask-task3-v0 | 0.02 (2/10000) | -16.0 | 0.0 | -8.003519 | 2.0014038 | -10360.0 | -2.0 | -4008.7666 | 2351.26 |
| puzzle-4x4-play-singletask-task4-v0 | 0.04 (4/10000) | -16.0 | 0.0 | -7.999373 | 1.995151 | -10055.0 | -1.0 | -4001.8894 | 2349.9917 |
| puzzle-4x4-play-singletask-task5-v0 | 0.04 (4/10000) | -16.0 | 0.0 | -7.999373 | 1.995151 | -10055.0 | -1.0 | -4001.8894 | 2349.9917 |
| cube-quadruple-singletask-task1-v0 | 0.001 (1/100000) | -4.0 | 0.0 | -3.901978 | 0.30897102 | -4000.0 | -1.0 | -1954.3688 | 1134.5515 |
| cube-quadruple-singletask-task2-v0 | 0.002 (2/100000) | -4.0 | 0.0 | -3.8990514 | 0.31413057 | -4000.0 | -1.0 | -1952.7515 | 1133.8807 |
| cube-quadruple-singletask-task3-v0 | 0.001 (1/100000) | -4.0 | 0.0 | -3.914329 | 0.28973544 | -4000.0 | -1.0 | -1960.2894 | 1137.1301 |
| cube-quadruple-singletask-task4-v0 | 0.0 (0/100000) | -4.0 | -1.0 | -3.9144847 | 0.28865227 | -4000.0 | -1.0 | -1960.2513 | 1137.1667 |
| cube-quadruple-singletask-task5-v0 | 0.012 (12/100000) | -4.0 | 0.0 | -3.968212 | 0.18561256 | -4000.0 | 0.0 | -1986.1152 | 1148.1842 |

## G   FULL EXPERIMENTAL RESULTS

We include the full experimental results in OGBench experiments in Table 9.

Table 9: Full offline RL Results in **30** OGBench tasks. $^*$ indicates the default task in each environment. We report the success rate (%) and 95% stratified bootstrap confidence interval over 4 runs.

| Task | #Data | FQL | N-step FQL | QC-FQL | CQN-AS | **DEAS** |
|---|---|---|---|---|---|---|
| scene-play-singletask-task1-v0 | | **100** $\pm 0$ | **100** $\pm 0$ | 99 $\pm 0$ | 2 $\pm 1$ | **99** $\pm 1$ |
| scene-play-singletask-task2-v0 | | 50 $\pm 7$ | 4 $\pm 3$ | 99 $\pm 1$ | 1 $\pm 1$ | **97** $\pm 1$ |
| scene-play-singletask-task3-v0 | 1M | **95** $\pm 2$ | 78 $\pm 5$ | 64 $\pm 8$ | 0 $\pm 0$ | 75 $\pm 6$ |
| scene-play-singletask-task4-v0$^*$ | | 3 $\pm 2$ | 0 $\pm 0$ | **68** $\pm 1$ | 0 $\pm 0$ | 65 $\pm 5$ |
| scene-play-singletask-task5-v0 | | 0 $\pm 0$ | 0 $\pm 0$ | 35 $\pm 7$ | 0 $\pm 0$ | **45** $\pm 6$ |
| cube-double-play-singletask-task1-v0 | | 46 $\pm 4$ | 17 $\pm 3$ | 68 $\pm 4$ | 7 $\pm 1$ | **76** $\pm 3$ |
| cube-double-play-singletask-task2-v0$^*$ | | 10 $\pm 2$ | 1 $\pm 0$ | 47 $\pm 8$ | 1 $\pm 1$ | **51** $\pm 8$ |
| cube-double-play-singletask-task3-v0 | 1M | 9 $\pm 2$ | 1 $\pm 1$ | 40$\pm 6$ | 0 $\pm 1$ | **47** $\pm 4$ |
| cube-double-play-singletask-task4-v0 | | 1 $\pm 1$ | 0 $\pm 0$ | 8$\pm 1$ | 1 $\pm 1$ | **8** $\pm 1$ |
| cube-double-play-singletask-task5-v0 | | 2 $\pm 1$ | 3 $\pm 1$ | 44$\pm 3$ | 0 $\pm 0$ | **57** $\pm 3$ |
| puzzle-3x3-play-singletask-task1-v0 | | **100** $\pm 0$ | 89 $\pm 3$ | 97 $\pm 1$ | 1 $\pm 2$ | **100** $\pm 0$ |
| puzzle-3x3-play-singletask-task2-v0 | | 19 $\pm 4$ | 40 $\pm 10$ | 81 $\pm 12$ | 0 $\pm 0$ | **94** $\pm 5$ |
| puzzle-3x3-play-singletask-task3-v0 | 1M | 15 $\pm 2$ | 14 $\pm 3$ | 50 $\pm 11$ | 0 $\pm 0$ | **91** $\pm 3$ |
| puzzle-3x3-play-singletask-task4-v0$^*$ | | 35 $\pm 4$ | 23 $\pm 3$ | 31 $\pm 4$ | 0 $\pm 0$ | **91** $\pm 3$ |
| puzzle-3x3-play-singletask-task5-v0 | | 47 $\pm 4$ | 13 $\pm 3$ | 50 $\pm 11$ | 0 $\pm 0$ | **96** $\pm 2$ |
| cube-triple-play-singletask-task1-v0 | | 31 $\pm 14$ | 17 $\pm 5$ | **100** $\pm 0$ | 0 $\pm 0$ | 98 $\pm 1$ |
| cube-triple-play-singletask-task2-v0$^*$ | | 9 $\pm 3$ | **91** $\pm 4$ | 92 $\pm 2$ | 0 $\pm 0$ | **95** $\pm 2$ |
| cube-triple-play-singletask-task3-v0 | 10M | 12 $\pm 5$ | 0 $\pm 0$ | 92 $\pm 2$ | 0 $\pm 0$ | **88** $\pm 3$ |
| cube-triple-play-singletask-task4-v0 | | 0 $\pm 1$ | 0 $\pm 0$ | **59** $\pm 7$ | 0 $\pm 0$ | 45 $\pm 7$ |
| cube-triple-play-singletask-task5-v0 | | 2 $\pm 1$ | 0 $\pm 0$ | 74 $\pm 4$ | 0 $\pm 0$ | **87** $\pm 5$ |
| puzzle-4x4-play-singletask-task1-v0 | | 54 $\pm 4$ | 28 $\pm 5$ | 66 $\pm 17$ | 0 $\pm 0$ | **92** $\pm 8$ |
| puzzle-4x4-play-singletask-task2-v0 | | 24 $\pm 3$ | 2 $\pm 1$ | **80** $\pm 16$ | 0 $\pm 0$ | 42 $\pm 7$ |
| puzzle-4x4-play-singletask-task3-v0 | 10M | 36 $\pm 4$ | 42 $\pm 7$ | 69 $\pm 22$ | 0 $\pm 0$ | **99** $\pm 1$ |
| puzzle-4x4-play-singletask-task4-v0$^*$ | | 22 $\pm 2$ | 28 $\pm 3$ | 70 $\pm 17$ | 0 $\pm 0$ | **88** $\pm 4$ |
| puzzle-4x4-play-singletask-task5-v0 | | 22 $\pm 4$ | 3$\pm 2$ | 61$\pm 19$ | 0 $\pm 0$ | **89** $\pm 6$ |
| cube-quadruple-play-singletask-task1-v0 | | 79 $\pm 6$ | 70 $\pm 9$ | 79 $\pm 7$ | 0 $\pm 0$ | **92** $\pm 5$ |
| cube-quadruple-play-singletask-task2-v0$^*$ | | 0 $\pm 0$ | **97** $\pm 2$ | 63 $\pm 7$ | 0 $\pm 0$ | **100** $\pm 0$ |
| cube-quadruple-play-singletask-task3-v0 | 100M | 6$\pm 3$ | 1 $\pm 1$ | 33 $\pm 7$ | 0 $\pm 0$ | **62** $\pm 9$ |
| cube-quadruple-play-singletask-task4-v0 | | 0 $\pm 0$ | 13 $\pm 5$ | **38** $\pm 7$ | 0 $\pm 0$ | 31 $\pm 7$ |
| cube-quadruple-play-singletask-task5-v0 | | 0 $\pm 0$ | 0 $\pm 0$ | 12 $\pm 6$ | 0 $\pm 0$ | **35** $\pm 10$ |

