# OpenReview forum: "DEAS: DEtached value learning with Action Sequence for Scalable Offline RL"
_ICLR.cc/2026/Conference — ICLR 2026 Poster_

### Official Review · Reviewer_dtkT · 2025-10-25

**Soundness:** 3
**Presentation:** 3
**Contribution:** 3
**Rating:** 8
**Confidence:** 3

**Summary:**

This paper proposes a value learning framework that leverages action sequences for value learning to facilitate complex, long-horizon sequential decision making. Experimental results on OGBench, RoboCasa Kitchen, and real-world tasks demonstrate the effectiveness of the proposed method.

**Strengths:**

- The motivation of the proposed method is well explained, and this paper is easy to follow.
- The proposed DEtached value learning with Action Sequence (DEAS) framework that decouples critic training from the actor, encouraging the value function to prioritize high-return actions observed in the offline dataset.
- The proposed approach outperforms baselines in various benchmarks and real-world tasks.

**Weaknesses:**

- No quantitative results are provided to demonstrate the prevention of value overestimation.
- The VLA experiments are conducted with only a single baseline. It would be more convincing to include comparisons with other VLA methods, such as OpenVLA [1].
- No sensitivity analysis is conducted on the hyperparameter $H$.

Reference:

[1] Kim et al. "OpenVLA: An Open-Source Vision-Language-Action Model", arXiv preprint arXiv:2406.09246, 2024.

**Questions:**

Have you compared your method with other competitive baselines such as CQL [1]?

Reference:

[1] Kumar et al. "Conservative Q-Learning for Ofﬂine Reinforcement Learning", NeurIPS, 2020.

---

> ### Author Response · Authors · 2025-11-20
>
> Dear Reviewer dtkT,
>
> We sincerely appreciate your valuable comments, which were extremely helpful in improving our draft. Below, we address each comment in detail.
>
> ---
> **[W1] Quantitative analyses on Q-value overestimations**
>
> Thank you for your valuable suggestions. To prove this, we analyze value calibration between critic outputs and true discounted returns, which holds that a well-trained critic should maintain a monotonic relationship between predicted Q values and actual returns across unseen state-action distributions. We compute critic outputs $\hat{Q}(s_t, a_{t:t+H-1})$ and discounted returns $\hat{G}(s_t, a_{t:t+H-1})$ using 5000 unseen trajectories, then partition transitions into bins by $\hat{Q}$ values and plot the average $\hat{G}$ within each bin, where deviation from the diagonal $y=x$ quantifies overestimation (above) or underestimation (below). As shown in Figure 5 of the original draft, DEAS exhibits a calibration curve consistently closer to the diagonal than QC-FQL, indicating significantly reduced overestimation and improved value alignment.
>
> **[W2] Comparison with additional baselines for VLA experiments**
>
> We agree that the choice of VLA backbone matters and explain our selection here. In our preliminary study, we compared two widely used and strong recent VLA models, $\pi_0$ [1] and GR00T N1.5 [2], both of which report higher performance than OpenVLA on standard benchmarks. Using 100 expert demonstrations, we observe that $\pi_0$ and GR00T N1.5 exhibit similar overall performance; we choose GR00T N1.5 as the backbone because its implementation makes experimentation in our pipeline easier. We added the updated results with $\pi_0$ baseline in the revised draft.
>
> **[W3] Clarification on sensitivity analyses over horizon $H$**
>
> Thank you for pointing this out. We would like to politely note that we conducted experiments on the effect of action sequence length $H$ in Section 5.3 of the original draft. Overall, we observe that DEAS fails to achieve meaningful performance with single- or two-step actions; performance improves with longer sequences, and it then requires proportionally larger actor networks to manage the increased action dimensions as the sequence lengthens.
>
> **[Q1] Comparison with other offline RL baselines**
>
> For choosing a representative baseline in OGBench experiments, we choose FQL [3] for two reasons: it significantly outperforms other offline RL baselines (which typically use Gaussian policy classes) by leveraging an expressive flow policy. Furthermore, as our paper focuses on investigating the effectiveness of value learning, we aim to use the same policy extraction schemes across methods, whereas comparing with other algorithms using Gaussian policy classes can violate this assumption.
>
> **Reference**\
> [1] Pi-0: A Vision-Language-Action Flow Model for General Robot Control, RSS 2025.\
> [2] GR00T N1.5: An Improved Open Foundation Model for Generalist Humanoid Robots, https://research.nvidia.com/labs/gear/gr00t-n1_5/ \
> [3] Flow Q-Learning, ICML 2025.

---

> > ### Comment · Reviewer_dtkT · 2025-11-21
> >
> > Thank you for your responses. I will maintain my original rating.

---

> ### Author Response · Authors · 2025-11-21
>
> Thank you for your feedback and valuable suggestions. Please feel free to reach out with any additional questions or suggestions.
>
> Thank you once again!
>
> Many thanks,\
> Authors

---

### Official Review · Reviewer_uCyh · 2025-10-28

**Soundness:** 3
**Presentation:** 3
**Contribution:** 3
**Rating:** 6
**Confidence:** 3

**Summary:**

The paper proposes DEAS, a method for offline reinforcement learning that models each decision as an action sequence rather than a single action. Building on the IQL framework, DEAS learns a critic consisting of $Q$ and $V$ networks while allowing an arbitrary policy architecture for action generation. During critic training, both the predicted and target values are treated as distributions and optimized via a cross-entropy loss. Experiments on continuous-control tasks, including both simulated and real-world robotic environments, demonstrate the effectiveness of the approach.

**Strengths:**

1. Effectively leverages action sequences to address sparse rewards and long-horizon credit assignment challenges.
2. Appropriately builds on the IQL framework for stable critic learning under the offline setting.
3. Demonstrates strong empirical performance on complex, long-horizon continuous-control tasks in both simulation and real-world environments.

**Weaknesses:**

1. The novelty of the paper is somewhat limited. It mainly combines existing ideas from action-sequence RL, IQL, and HLG.
2. The cross-entropy loss formulations in Equation (1) and Equation (2) are confusing. Typically, the predicted distribution appears inside the logarithm term and the target distribution outside. In contrast, both equations place the target inside the log term, which is unconventional. Moreover, in Equation (2), the role of parameter $\theta$ on the right-hand side is unclear, as only $\psi$ and $\hat{\theta}$ are explicitly shown.
3. The paper provides limited discussion on how to learn the policy based on the trained critic. Since the policy outputs an entire sequence of actions per decision, the resulting product action space becomes extremely large, making policy learning challenging. However, the authors do not explain how this issue is addressed.

**Questions:**

1. How can the policy be effectively learned when each decision involves generating a high-dimensional action sequence? Does the paper propose any mechanism to mitigate the exponential growth of the action space?
2. Why do Equations (1) and (2) place the target distribution inside the logarithm term in the cross-entropy loss, and how should we interpret the missing parameter $\theta$ in Equation (2)?
3. How are the support bounds $v_{min}$ and $v_{max}$determined? These values appear to constrain the expressiveness of the critic network, while setting a larger range may make the learning process more difficult.

Minor comments:
1. The description of IQL in Section 3 is inaccurate. The paper states that “By using $\tau > 0.5$, Equation 3 penalizes the overestimated value in out-of-distribution actions.” (By the way, there is no Equation 3.) However, since $u = Q - V$, where $Q$ is obtained from the target network and $V$ is the learnable network, the equation should instead penalize the **underestimated** values of the value network.
2. There is also no Equation (4.2), as referenced in Algorithm 1. Please verify and correct the numbering.
3. It is recommended to mention HLG again after Equation (2).

---

> ### Author Response · Authors · 2025-11-20
>
> Dear Reviewer uCyh,
>
> We sincerely appreciate your valuable comments, which were extremely helpful in improving our draft. Below, we address each comment in detail.
>
> ---
>
> **[W1] Clarifying the novelty aspect**
>
> We appreciate the reviewer’s concern. We would like to emphasize that DEAS is not a simple integration of existing components. Our design is driven by specific failure modes in offline RL with action sequences: naive actor–critic training leads to value overestimation, multi-step targets amplify bootstrap bias, and stability degrades as horizons and model scale increase. This motivated our particular combination of (i) sequence-level detached value learning to remove unstable actor–critic coupling, (ii) distributional value learning on top of an expectile objective to control bootstrap bias, and (iii) a dual-discount scheme to tune within-sequence and across-sequence credit assignment separately.
> We empirically evaluated several alternative integration choices (e.g., standard actor–critic, non-distributional, and single-discount variants) across different tasks and model setups. We found that the proposed configuration is consistently the most stable and effective. In the revised draft, we also add calibration-based analyses on held-out data, underscoring that our design substantially mitigates overestimation.
>
> **[W2, Q2] Clarifying notations**
>
> Thank you for pointing this out. We would like to politely note that we denote the predicted probabilities with $\hat{p}$ and the target probabilities with $p$ following [1], and properly place the predicted probabilities inside the logarithm when defining the cross-entropy loss. We further clarify this in the revised draft for better visibility. In Equation (2) of the original paper, the predicted probability from the Q-function should be parameterized by $\theta$. We fixed this typo in the revised draft and colored it in cyan.
>
> **[W3, Q1] Discussion on training policies with increased action spaces**
>
> For handling high-dimensional sequence action spaces, we use an expressive flow-based policy that naturally generates coherent action sequences [2, 3]. Our VLA experiments show that even with $H=16$, a large-scale flow-matching VLA can be trained stably under DEAS. In this work, our primary focus is on establishing a stable and effective recipe for value learning with action sequences, rather than proposing advanced mechanisms for policy optimization in exponentially large action spaces. Extending our approach to much longer horizons with explicit hierarchical or option-based sequence generation is an interesting direction for future work, which we will clarify in the revised draft.
>
> **[Q2] Rules for determining support bounds $\mathbf{v}_{min}$/$\mathbf{v}_{max}$**
>
> We would like to mention that we provided the rules for determining support bounds used in distributional RL in Appendix A.1 of the original draft. First, we calculate the discounted returns for the offline dataset and select the 1% and 99% percentiles as the lower and upper bounds. Additionally, we compute the theoretical upper and lower bounds of the return using the possible reward range from the environment, horizon $L$, action sequence length $H$, and discount factors $gamma_1$ and $gamma_2$. We then choose the procedure with the best performance on the default task in each environment.
>
> **[C1, C2, C3] Fixing descriptions and notations in the draft**
>
> We thank the reviewer for raising this point. To clarify the expectile objective: with (u = \bar{Q}(s, a) - V(s)) and (L_2^\tau(u) = \lvert \tau - \mathbf{1}(u < 0)\rvert u^2), choosing (\tau > 0.5) places greater weight on cases where (V) underestimates (\bar{Q}) ((u>0)) than on overestimation, so (V) is guided toward an upper expectile of the TD targets. This mechanism does not directly “penalize overestimated values,” and we updated the manuscript to reflect this more precise interpretation and remove the incorrect “Equation 3” reference. In accordance with Equation (4.2) and Algorithm 1, we applied the numbering and updated the references in the revised draft. These updates are temporarily highlighted in "cyan" for your convenience.
>
> **Reference**\
> [1] Stop Regressing: Training Value Functions via Classification for Scalable Deep RL, ICML 2024.\
> [2]  Pi-0: A Vision-Language-Action Flow Model for General Robot Control, RSS 2025.\
> [3] GR00T N1.5: An Improved Open Foundation Model for Generalist Humanoid Robots, https://research.nvidia.com/labs/gear/gr00t-n1_5/

---

> ### Author Response · Authors · 2025-11-26
> **Gentle Reminder**
>
> Dear Reviewer uCyh,
>
> Thank you once again for your time and thoughtful review of our paper.
>
> As the discussion period is about to conclude, we would like to remind you that if you have any remaining comments, please share them. We believe that we have sincerely and successfully addressed your concerns, supported by the corresponding additional experimental results.
>
> If you have any further concerns or questions, please feel free to let us know.
>
> Thank you very much,\
> Authors

---

> > ### Author Response · Authors · 2025-12-02
> > **Additional Experiments with longer horizon $H$**
> >
> > Dear Reviewer uCyh,
> >
> > Thank you again for your valuable time and insightful review of our paper.
> >
> > We appreciate your suggestions in W3 and Q1. Following this, we conducted additional experiments, applying DEAS with $\pi_0$ [1], a Vision-Language Agent (VLA) with a significantly longer action sequence of $H=50$. The results, presented in the table below, demonstrate a clear improvement over the baseline methods.
> >
> > This outcome strongly confirms that our method reliably enhances VLAs, regardless of the chosen backbone or the length of the action sequence. We have updated Section 5.2.1 of the draft to include this new result, highlighted in $\text{\color{cyan}{cyan}}$.
> >
> > We are grateful for your helpful comments and suggestions!
> >
> > Authors.
> >
> > \begin{array}{lccccc}
> > \hline
> > \text{Models} &
> > \text{CoffeeSetupMug} &
> > \text{PnPC2M} &
> > \text{PnPM2C} &
> > \text{TurnOffStove} &
> > \text{Avg.} \newline
> > \hline
> > \pi_0^{\dagger} &
> > 20.0 &
> > 11.3 &
> > 10.0 &
> > 8.0 &
> > 12.3 \newline
> > \quad +\ \text{Filtered BC} &
> > \underline{30.7} &
> > \mathbf{16.7} &
> > \underline{14.7} &
> > \underline{10.0} &
> > \underline{18.0} \newline
> > \quad +\ \mathbf{\text{DEAS (Ours)}} &
> > \mathbf{37.3} &
> > \underline{15.3} &
> > \mathbf{19.3} &
> > \mathbf{15.3} &
> > \mathbf{21.8} \newline
> > \hline
> > \end{array}
> >
> > **Reference**\
> > [1] Pi-0: A Vision-Language-Action Flow Model for General Robot Control, RSS 2025.

---

### Official Review · Reviewer_EcDt · 2025-10-28

**Soundness:** 3
**Presentation:** 3
**Contribution:** 2
**Rating:** 2
**Confidence:** 4

**Summary:**

The paper addresses the challenge of scaling offline reinforcement learning to complex, long-horizon tasks. The authors identify that while using action sequences (instead of single actions) can reduce the effective planning horizon, it introduces severe value overestimation when combined with standard actor-critic frameworks.The core contribution is DEAS (DEtached value learning with Action Sequence), a method that learns a value function over $H$-step action sequences. To prevent value overestimation, DEAS "detaches" the critic and value function training from the actor. It does this by adopting an IQL-style expectile regression loss, which learns the value function by regressing only on the in-sample action sequences found in the offline dataset. This avoids querying the actor for out-of-distribution actions.The framework is stabilized by incorporating distributional RL and dual discount factors ($\gamma_1$ for intra-option rewards, $\gamma_2$ for inter-option values). The authors demonstrate that DEAS significantly outperforms existing single-step and action-sequence baselines on complex OGBench tasks and can be used to improve the performance of large-scale Vision-Language-Action (VLA) models in simulation and on a real robot.

**Strengths:**

1.  **Clear Problem Identification:** The paper clearly articulates a significant and practical problem: coupled actor-critic methods are unstable when the action space is $\mathcal{A}^H$.
2.  **Strong Empirical Results:** This is the paper's greatest strength. DEAS demonstrates state-of-the-art performance, with significant gains over baselines on the challenging OGBench tasks.
3.  **Real-World & VLA Validation:** The authors successfully scale the method to large VLA models and demonstrate its effectiveness and stability on a real robot. The fact that the QC baseline *collapses* on the real-world task while DEAS improves performance is a very strong supporting data point.
4.  **Effective Ablations:** The ablation studies in Table 4 are well-executed and convincingly demonstrate the necessity of each component of the method (e.g., $H > 2$, the combination of IQL and distributional RL).

**Weaknesses:**

1.  **Limited Conceptual Novelty:** As stated in the *Contribution* section, the method is a synthesis of IQL, fixed-duration options, and distributional RL. The contribution is an empirical and engineering one, not a fundamental algorithmic advance.
2.  **Imprecise Technical Terminology:** The paper repeatedly invokes "SMDP Q-learning" and the "options framework", but the method itself is not a true SMDP. The use of a *fixed, deterministic* duration $H$ does not create an option induced Semi-Markov process. The system is, more accurately, a standard 1st-order MDP where the agent selects from a temporally-extended, high-dimensional action space $\mathcal{O} = \mathcal{A}^H$. This imprecise language is confusing and oversells the connection to the general options framework. From this point of view, it's only another variant of ordinary chunked action sequences and the performance boost whether come from temporal abstraction is skeptical.
3. **Unjustified Optimality**: Detaching value functions and fixed horizon clearly alters the underlying decision process compares original SMDP. The convergence and optimality of DEAS is unjustified and not bounded.
4.  **Poor Scalability with Horizon $H$:** The paper's own ablation study reveals a significant scalability limitation. The authors state that "when the sequence length becomes longer than 8, it requires proportionally larger actor networks to handle the increased action dimensions". This creates a direct and problematic coupling between the task horizon ($H$) and the required network capacity, which severely limits the method's applicability to *truly* long-horizon problems.
5.  **Increased Hyperparameter Sensitivity:** The method adds several critical new hyperparameters that require tuning, including the sequence length $H$ (which Table 4a shows is task-dependent), and the dual discount factors $\gamma_1$ and $\gamma_2$, which the authors note are "critical for stable training".

**Questions:**

1.  Can the authors clarify the novelty of the method beyond the combination of IQL and action sequences? Is the primary contribution the empirical demonstration that this combination is effective, or is there a more fundamental insight that I have missed?
2.  Regarding the scalability weakness: The paper notes the actor network must scale with $H$. This seems to be a major bottleneck. Have the authors investigated methods to decouple this than the entire $H$-step block at once?
3.  Given that the duration $H$ is fixed and deterministic, why was the method presented as an SMDP rather than, more precisely, as a standard MDP with a temporally-extended action space $\mathcal{O} = \mathcal{A}^H$? I suggest authors give this paper a major re-write and detach concepts such as SMDPs and options out of scope.
4.  Given the engineering success and strong empirical performance, I am happy to raise score if authros position their contribution from a more rigorous view.

---

> ### Author Response · Authors · 2025-11-20
>
> Dear Reviewer EcDt,
>
> We sincerely appreciate your valuable comments, which were extremely helpful in improving our draft. Below, we address each comment in detail.
>
> ---
>
> **[W1, Q1] Clarification on novelty**
>
>
> Thank you for your point. We would like to emphasize that we do not claim a fundamentally new RL paradigm; instead, DEAS is driven by specific failure modes in offline RL with action sequences: naive actor–critic training leads to value overestimation, multi-step targets amplify bootstrap bias, and stability degrades as horizons and model scale increase. This motivated our particular combination of (i) sequence-level detached value learning to remove unstable actor–critic coupling, (ii) distributional value learning on top of an expectile objective to control bootstrap bias, and (iii) a dual-discount scheme to tune within-sequence and across-sequence credit assignment separately. In addition to this design, we demonstrate the practical viability of offline RL with a large flow-based VLA across diverse long-horizon tasks. We also include quantitative calibration analyses in the revised draft, showing that DEAS effectively mitigates value overestimation across unseen offline datasets.
>
>
> **[W2, Q2] Additional explanation on terminologies**
>
> We appreciate your valuable comments. DEAS can be interpreted as a **special case** of the options/SMDP perspective, but we agree that our earlier wording somewhat overstated this connection. Consequently, we have revised the draft to position DEAS primarily as an extension of Q-learning that incorporates an extended action space, and we now present the options/SMDP perspective as an optional interpretation in the Appendix, rather than a core assertion. For clarity, all modified sections have been highlighted in cyan.
>
> **[W3] Questions on Optimality**
>
> Thank you for pointing this out. Following your suggestion, we provide the proof on the convergence and optimality of DEAS in Appendix E of the revised draft. In a nutshell, we prove that Theorem 3 of the original IQL paper [1]—including Lemma 2 and Corollary 2.1—extends directly to this sequence-action setting without modification, by treating the action sequence as a single macro-action.
>
> **[W4, Q3] Scalability with horizon $H$**
>
> Thank you for the valuable feedback. Our VLA experiments already show that, even with (H=16), a large-scale flow-matching VLA can be trained stably under DEAS, suggesting that pretrained high-capacity models can effectively absorb the increased dimensionality of more extended action sequences. In addition, we are conducting further experiments on ($\pi_0$ [2]) with substantially longer sequences (H=50) to examine horizon scalability more directly, and we plan to share these results during the discussion phase. Extending DEAS to even longer horizons through hierarchical or option-based sequence generation remains a promising direction, and we clarified this limitation and future path in the revised draft.
>
> **[W5] Discussion on hyperparameter sensitivity**
>
>
> Thank you for pointing this out. Regarding the dual discount factor, we observe that the main factor is using a lower value of $tau_1$, and the result aligns with $tau_2$. Concerning the action sequence $H$, we found that either 4 or 8 works well for most tasks, and even larger action sequences can be suitable when the model size is sufficiently large, such as in the VLA scale. We also demonstrate the possibility of using publicly released large-scale models.
>
> **Reference**\
> [1] Offline Reinforcement Learning with Implicit Q-Learning, ICLR 2022.\
> [2] Pi-0: A Vision-Language-Action Flow Model for General Robot Control, RSS 2025.

---

> > ### Comment · Reviewer_EcDt · 2025-11-21
> > **reject for overselling**
> >
> > I have read the authors' response. However, the insistence (in main text and algorithm) that this method represents "learning options" or a "special case of SMDPs" prevents me from raising my score.
> >
> > The mathematical definition of an SMDP involves variable dependence lengths. In rigorous Reinforcement Learning literature, an Option is defined by a tuple $(I, \pi, \beta)$, where $\beta$ represents a stochastic termination condition. The complexity of SMDP learning arises specifically because the duration $\tau$ is a random variable dependent on the state trajectory. In DEAS, the duration is a fixed hyperparameter $k$. Describing a standard MDP with an extended action space $\mathcal{A}^k$ as a "special case of an SMDP" is mathematically trivial and terminologically misleading. Acknowledging it's a "special case" and essentially an MDP with a clever trick, feels like a deliberate misrepresentation. This constitutes "overselling" the conceptual novelty.
> >
> > DEAS uses a fixed horizon $k$. This is an augmented action sequence in a standard First-Order MDP, not a Semi-Markov Decision Process. To claim otherwise disregards the mathematical rigor required for definitions in this field.
> >
> > The strength of this paper lies in its empirical results, not in theoretical novelty regarding options. Please strictly re-scope the paper (including the Appendix) to reflect that this is an action-sequence method. I never recommend acceptance for a paper that fundamentally mischaracterizes its own theoretical basis to appear more novel.

---

> ### Author Response · Authors · 2025-11-21
>
> Dear Reviewer EcDt,
>
> Thanks for your response. We agree that, under a fixed, deterministic duration $H$, our method is more appropriately framed as a standard MDP with a temporally extended action space ($\mathcal{A}^H$), and that referring to SMDPs and options can be misleading or suggest an unwarranted connection to the general options framework. In response, we have completely removed SMDP- and options-related terminology from the paper—including the appendix—and uploaded a revised draft that consistently presents the method as an MDP with sequence actions (changes are highlighted in $\text{\color{magenta}magenta}$).
>
> We hope this more precise technical framing better reflects the scope of our contribution and addresses the reviewer’s concern. If you have any further questions or suggestions, please do not hesitate to let us know.
>
> Many thanks,\
> Authors

---

> ### Author Response · Authors · 2025-11-26
> **Gentle Reminder**
>
> Dear Reviewer EcDt,
>
> Thank you once again for your time and thoughtful review of our paper.
>
> As the discussion period is about to conclude, we would like to remind you that if you have any remaining comments, please share them. We believe we have sincerely and successfully addressed your concerns, supported by the additional experimental results and the carefully revised draft.
>
> If you have any further concerns or questions, please feel free to let us know.
>
> Thank you very much,\
> Authors

---

> ### Author Response · Authors · 2025-12-01
> **Additional Experiments with longer horizon $H$**
>
> Dear Reviewer EcDt,
>
> Thank you again for your valuable time and insightful review of our paper.
>
> We appreciate your suggestions in W4 and Q3. Following these, we conducted additional experiments, applying DEAS with $\pi_0$ [1], a Vision-Language Agent (VLA) using a significantly longer action sequence of $H=50$. The results, presented in the table below, demonstrate a clear improvement over the baseline methods.
>
> This outcome strongly confirms that our method reliably enhances VLAs, regardless of the chosen backbone or the length of the action sequence. We have updated Section 5.2.1 of the draft to include this new result, highlighted in $\text{\color{cyan}{cyan}}$.
>
> We are grateful for your helpful comments and suggestions!
>
> Authors.
>
> \begin{array}{lccccc}
> \hline
> \text{Models} &
> \text{CoffeeSetupMug} &
> \text{PnPC2M} &
> \text{PnPM2C} &
> \text{TurnOffStove} &
> \text{Avg.} \newline
> \hline
> \pi_0^{\dagger} &
> 20.0 &
> 11.3 &
> 10.0 &
> 8.0 &
> 12.3 \newline
> \quad +\ \text{Filtered BC} &
> \underline{30.7} &
> \mathbf{16.7} &
> \underline{14.7} &
> \underline{10.0} &
> \underline{18.0} \newline
> \quad +\ \mathbf{\text{DEAS (Ours)}} &
> \mathbf{37.3} &
> \underline{15.3} &
> \mathbf{19.3} &
> \mathbf{15.3} &
> \mathbf{21.8} \newline
> \hline
> \end{array}
>
> **Reference**\
> [1] Pi-0: A Vision-Language-Action Flow Model for General Robot Control, RSS 2025.

---

### Official Review · Reviewer_FBjU · 2025-11-04

**Soundness:** 3
**Presentation:** 3
**Contribution:** 2
**Rating:** 4
**Confidence:** 4

**Summary:**

This paper presents an offline RL framework that tackles the value-overestimation problem through detached value learning. To improve stability, it also integrates distributional RL and dual discount factors for intra- and inter-option transitions. Empirically, DEAS is shown to outperform baselines on 30 long-horizon OGBench tasks and to improve large Vision-Language-Action (VLA) systems on both RoboCasa simulations and real Franka robot manipulation tasks. The main contributions are: (1) framing sequence-level offline RL under SMDP Q-learning with detached critics, (2) a recipe (distributional RL + dual discounts) for stable learning with action sequences, and (3) demonstrating practical gains on long-horizon benchmarks and VLA applications.

**Strengths:**

1. The experiments applying DEAS to large VLA models are a strong selling point: DEAS improves VLA performance using mixed-quality offline rollouts plus a small set of expert data, which is important for real-world robotics.
2. The manuscript is structured logically and presents a well-targeted technical design.

**Weaknesses:**

1. My main concern is that the paper’s core components are largely integrations or adaptations of prior work, such as IQL-style detached learning, distributional RL methods, and option/sequence mechanisms similar to Q-Chunking.
2. Detached value learning updates the critic using in-distribution high-return actions, but when the offline dataset contains predominantly suboptimal trajectories, this strategy may still yield biased or miscalibrated value estimates (under- or over-pessimism). The paper does not adequately verify whether the learned value function aligns with true returns across different data quality levels. I recommend that the authors analyze Q-error across bins of trajectory quality and conduct sensitivity studies using datasets with controlled expert-to-suboptimal ratios (e.g., 10%, 30%, 50% expert data).
3. Several key equations are unnumbered and inconsistently referenced, which increases the difficulty of understanding Algorithm 1.
4. Although the central motivation of DEAS is to alleviate value overestimation induced by action sequences, the experimental results only report task-level success metrics without providing direct analyses of overestimation (e.g., predicted Q-values vs. realized returns). Including such quantitative evidence would substantially strengthen the paper’s claims.

**Questions:**

1. Do you have experiments integrating DEAS (action sequences + detached critic) with representative methods such as CQL? For example, does adding detached sequence-level critics to CQL further reduce overestimation or improve long-horizon performance?
2. To address potential value miscalibration, have you evaluated DEAS across datasets with varying proportions of expert trajectories (e.g., 10%/30%/50% expert)? How does data quality influence (a) final performance, and (b) the calibration of predicted Q vs realized returns? If absent, please add such sensitivity analysis or discuss limitations clearly.

---

> ### Author Response · Authors · 2025-11-20
>
> Dear Reviewer FBjU,
>
> We sincerely appreciate your valuable comments, which were extremely helpful in improving our draft. Below, we address each comment in detail.
>
> ---
>
> **[W1] Clarifying the novelty aspect**
>
> We appreciate the reviewer’s concern. We would like to emphasize that DEAS is not a simple integration of existing components. Our design is driven by specific failure modes in offline RL with action sequences: naive actor–critic training leads to value overestimation, multi-step targets amplify bootstrap bias, and stability degrades as horizons and model scale increase. This motivated our novel combination of (i) sequence-level detached value learning to remove unstable actor–critic coupling, (ii) distributional value learning on top of an expectile objective to control bootstrap bias, and (iii) a dual-discount scheme to tune within-sequence and across-sequence credit assignment separately.
> We empirically evaluated several alternative integration choices (e.g., standard actor–critic, non-distributional, and single-discount variants) across different tasks and model setups. We found that the proposed configuration is consistently the most stable and effective. In the revised draft, we also add calibration-based analyses on held-out data, underscoring that our design substantially mitigates overestimation.
>
> **[W4] Quantitative analyses on Q-value overestimations**
>
> Thank you for your valuable suggestions. To prove this, we analyze value calibration between critic outputs and true discounted returns, which holds that a well-trained critic should maintain a monotonic relationship between predicted Q values and actual returns across unseen state-action distributions. We compute critic outputs $\hat{Q}(s_t, a_{t:t+H-1})$ and discounted returns $\hat{G}(s_t, a_{t:t+H-1})$ using 5000 unseen trajectories. Then, we partition transitions into bins based on $\hat{Q}$ values and plot the average $\hat{G}$ within each bin, where deviation from the diagonal $y=x$ quantifies overestimation (above) or underestimation (below). As shown in Figure 5 of the revised draft, DEAS exhibits a calibration curve consistently closer to the diagonal than QC-FQL, indicating significantly reduced overestimation and improved value alignment.
>
>
> **[W2, Q2] Experiments on Q-errors with different data qualities**
>
> Firstly, OGBench comprises a wide range of return distributions: in all datasets, only a small fraction of trajectories achieve task success, and in some cases, there are no fully successful demonstrations at all (see Table 8 in the revised draft). Our observation is that DEAS can consistently outperform across all tasks, even in settings where no trajectory achieves complete success.
>
> Following your suggestion, we additionally conduct experiments on mixed datasets constructed by combining the $\tt{play}$ and $\tt{noisy}$ datasets from OGBench, where the $\tt{noisy}$ datasets contain transitions with substantially lower returns. As shown in Figure 6 of the revised draft, DEAS consistently outperforms QC-FQL, and its calibration curve stays closer to the diagonal across all regimes, indicating better calibration and further supporting the effectiveness of our method. Furthermore, we report success rates, where DEAS outperforms QC-FQL across all data regimes.
>
> **[W3] Omitted numbering in equations**
>
> Thank you for pointing this out. To resolve this issue, we applied the numbering and improved the reference in the revised draft. These updates are temporarily highlighted in "cyan" for your convenience.
>
> **[Q1] Integrating DEAS with CQL**
>
>
> Thanks for your suggestion. CQL [1] is an offline RL method that regularizes the learned Q-function by penalizing value estimates for out-of-distribution actions, thereby reducing overestimation and distributional shift. While CQL and DEAS share this goal, they do so differently: DEAS decouples value and policy to avoid extrapolation error, whereas CQL relies on policy-sampled actions during Q-updates, so combining them would mostly duplicate conservatism and reintroduce coupling between value and policy without a clear additional benefit.
>
> **References**\
> [1] Conservative Q-Learning for Offline Reinforcement Learning, NeurIPS 2020.

---

> > ### Comment · Reviewer_FBjU · 2025-11-22
> >
> > The authors’ response and the corresponding revisions have addressed most of my concerns. Although I am still not fully convinced by the novelty of the proposed method, the additional experiments on sub-optimal data and the further analysis of value overestimation make the paper more complete and technically sound in its revised form. Therefore, I am willing to increase my score.

---

> > > ### Author Response · Authors · 2025-11-22
> > >
> > > Thank you for your response.\
> > > We are happy to hear that we have addressed most of your concerns. Following your suggestion, we will further clarify the novelty of our method in the final manuscript. If you have any further questions or suggestions, please do not hesitate to let us know. Again, thank you for the valuable suggestion and your positive assessment of our work.
> > >
> > > Many thanks,\
> > > Authors

---

### Author Response · Authors · 2025-11-20
**General Response**

We deeply appreciate your time and effort in reviewing our manuscript. As the reviewers highlighted, DEAS is a simple yet effective offline RL framework that leverages action sequences for long-horizon, complex robotic manipulation tasks (all reviewers). Our paper clearly identifies the instability issue in coupled actor-critic methods for sequence-level action spaces (FBjU, EcDt), and proposes a stable, practical recipe (detached value learning, distributional networks, dual discount factor) for long-horizon offline RL (all reviewers) with clear presentations (FBjU, dtkT). DEAS exhibits strong empirical performance on complex tasks (all reviewers), scales effectively to large VLA models, and offers especially compelling real-world results (EcDt, dtkT, FBjU). Furthermore, we provide comprehensive ablation studies (EcDt, dtkT) to demonstrate the necessity and contribution of each component in our framework.

We appreciate the reviewers’ insightful comments on our manuscript. In response to the questions and concerns you raised, we have carefully revised and enhanced the manuscript with the following additional experiments and discussions:
- Fixing technological terminologies and clarifying contributions (Section 1, Section 4.1, Section 6)
- Improving the method description and fixing typos (Section 4.2)
- Comparison with $\pi_0$ baseline (Section 5.2.1)
- Additional experimental results (Section 5.3)
  - Systematically measuring alignment with real returns by analyzing value calibration between predicted critic values and real returns.
  - Examining robustness to dataset qualities by training agents with different data mixtures
- Additional discussion on limitations and future direction (Appendix A)
- Adding figures for specifying robot platform used in our expereiments (Appendix B)
- Providing theoretical proof of the optimality of DEAS (Appendix E)
- Providing statistics of datasets used in OGBench experiments (Appendix G)

These updates are temporarily highlighted in "cyan" for your convenience.
We strongly believe that DEAS can be a useful addition to the ICLR community, particularly because reviewers’ constructive comments enhanced the manuscript.
Thank you very much,
Authors

---

### Meta-Review · Area_Chair_AUWS · 2026-01-07

**Summary:**

The original scores were diverse with 8/6/4/2.
Reviewers broadly agreed that the paper addresses an important and practical problem and presents strong empirical results.
The application to large VLA models and the demonstrated robustness on real robots are considered major strengths.
The main concerns raised by reviewers center on limited methodological novelty, potential overselling or mispresent of the theoretical framing (SMDP/options), scalability with respect to sequence horizon, and clarity of technical presentation (notation, terminology, and equations).
During rebuttal, the authors provided additional analyses, including explicit longer-horizon evaluations, Q-value calibration curves, and controlled data-quality experiments. The mis-used SMDP/options terminology is removed. These revisions addressed many of the most serious concerns and strengthened the paper’s technical rigor.

While the contribution is not a fundamentally new RL paradigm, the combination of careful design choices, strong empirical validation, and demonstrated real-world applicability makes the paper a solid contribution to offline RL and long-horizon decision making.

Considering the overall balance of resolved and remaining concerns, as well as the reviewers’ follow-up responses, the AC recommends acceptance and suggests that the authors carefully revise the final version to clearly reflect the empirical nature of the contribution and incorporate the clarifications and analyses introduced during the rebuttal.

**Reviewer Concerns:**

During the rebuttal, several concerns were addressed, including:
- Value overestimation and calibration (FBjU, dtkT). The authors added explicit calibration analyses comparing predicted Q-values with realized returns on unseen trajectories and under varying data-quality regimes.


- Sensitivity to data quality (FBjU). Additional experiments under different data-quality mixtures were provided.


- Scalability to longer horizons (EcDt, uCyh). The authors added experiments with H=50, which partially alleviates concerns about horizon scalability.


- Misuse / overclaiming of SMDP/options terminology (EcDt). This was a major issue. The initial rebuttal did not fully resolve the concern and prevented the reviewer from raising their score. Subsequently, the authors explicitly acknowledged the issue and removed SMDP/options terminology from both the main text and appendix.


Some concerns still remain. These are primarily about scope and framing rather than correctness, including:
- Limited novelty (all reviewers). While the revised positioning is improved, the novelty concern remains: DEAS is still best viewed as a careful synthesis and stabilization of existing components. That said, given the strong evidence on large-scale models and real robotic applications, the core contribution is better framed as a empirical and engineering advance, rather than a fundamentally new algorithmic framework.


- Theoretical optimality (EcDt). Although convergence arguments were added, the broader theoretical implications of fixed-horizon action-sequence formulations remain limited.

**Reviewer Scores:**

- Reviewer FBjU. Explicitly stated that the rebuttal addressed most concerns and indicated willingness to increase the score, likely from 4 to 6.


- Reviewer EcDt. The most critical (score 2) due to overselling and terminology misuse. The first round of rebuttal didn’t fully resolve the question, avoiding the reviewer to increase the score. After the authors fully removed SMDP/options framing, their stated condition for reconsideration might be met; a reasonable expectation would be an increase to 4 (or with some possibility to 6), though still cautious on novelty.


- Reviewer uCyh. Maintain 6. Most technical concerns were addressed; novelty concerns remain but are not disqualifying.


- Reviewer dtkT. Maintain 8. Concerns on overestimation analysis and baselines were satisfactorily addressed.

---

### Decision · Program_Chairs · 2026-01-26

Accept (Poster)